# Lamina-specific AMPA receptor dynamics following visual deprivation in vivo

Han L Tan[1], Richard H Roth[1], Austin R Graves[1], Robert H Cudmore[2], Richard L Huganir[1]*

[1]Solomon H Snyder Department of Neuroscience and Kavli Neuroscience Discovery Institute, Johns Hopkins University School of Medicine, Baltimore, United States; [2]Department of Physiology and Membrane Biology, University of California School of Medicine, Davis, United States

**Abstract** Regulation of AMPA receptor (AMPAR) expression is central to synaptic plasticity and brain function, but how these changes occur in vivo remains elusive. Here, we developed a method to longitudinally monitor the expression of synaptic AMPARs across multiple cortical layers in awake mice using two-photon imaging. We observed that baseline AMPAR expression in individual spines is highly dynamic with more dynamics in primary visual cortex (V1) layer 2/3 (L2/3) neurons than V1 L5 neurons. Visual deprivation through binocular enucleation induces a synapse-specific and depth-dependent change of synaptic AMPARs in V1 L2/3 neurons, wherein deep synapses are potentiated more than superficial synapses. The increase is specific to L2/3 neurons and absent on apical dendrites of L5 neurons, and is dependent on expression of the AMPAR-binding protein GRIP1. Our study demonstrates that specific neuronal connections, across cortical layers and even within individual neurons, respond uniquely to changes in sensory experience.

*For correspondence:
rhuganir@jhmi.edu

Competing interests: The authors declare that no competing interests exist.

## Introduction

Neuronal circuits in the brain are subject to synaptic plasticity mechanisms induced by sensory experience (*Ko et al., 2013*) and learning (*Chen et al., 2015*; *Peters et al., 2017*) while exhibiting a critical ability to maintain network activity within a normal operating range during perturbations such as sensory deprivation (*Hengen et al., 2016*; *Turrigiano, 2012*). The homeostatic regulation of neuronal activity has been demonstrated in vivo, where chronic visual deprivation through monocular eyelid suture induces an initial decline in activity of pyramidal neurons in the visual cortex that is eventually restored to baseline (*Hengen et al., 2013*; *Hengen et al., 2016*). Other studies using enucleation to deprive the vision show that the recovery of neuronal activity is accompanied by an increase in spine size (*Barnes et al., 2015*; *Keck et al., 2013*). However, the molecular mechanisms underlying this homeostatic regulation of neuronal activity in vivo have not been much investigated. Further, whether these homeostatic mechanisms occur homogenously across the individual neuron or are specific to individual dendritic compartments remains elusive.

AMPA receptors (AMPARs) are the principle postsynaptic glutamate receptors mediating fast excitatory synaptic transmission, and regulation of AMPAR trafficking is critical for synaptic plasticity and brain function (*Diering and Huganir, 2018*; *Volk et al., 2015*). One of the major forms of homeostatic regulation of neuronal activity involves the modulation of AMPAR expression at synapses and this has been extensively characterized in vitro and ex vivo (*Desai et al., 2002*; *Goel et al., 2006*; *Goel and Lee, 2007*; *O'Brien et al., 1998*; *Turrigiano, 2012*; *Turrigiano et al., 1998*), where chronic visual deprivation induces up-regulation of synaptic AMPARs in the primary visual cortex (V1). However, whether sensory-deprivation-induced homeostatic regulation of AMPAR trafficking occurs in vivo is unknown.

Here, by using longitudinal in vivo two-photon imaging of fluorescently labeled synaptic AMPAR expression, which is a good proxy for postsynaptic strength, we sought to investigate the underlying molecular mechanisms of homeostatic plasticity in vivo and examine how sensory deprivation affects real-time AMPAR dynamics with single-synapse resolution in awake, unanesthetized animals. We observed that AMPAR expression within individual spines in V1 is highly dynamic under normal conditions and that this dynamic is cell type-specific and visual experience-dependent. Visual deprivation by binocular enucleation initially decreased synaptic AMPARs on apical dendrites of V1 layer 2/ 3 (L2/3) neurons, but the expression then recovered and subsequently underwent further increase with prolonged deprivation. The later increase of AMPARs induced by deprivation is absent on apical dendrites of L5 neurons, indicating that the homeostatic regulation of AMPARs is cell type-specific. Further, within L2/3 neurons, the increases of synaptic AMPARs on basal dendrites following visual deprivation are earlier and larger than on apical dendrites, suggesting a depth-dependent mechanism. Finally, we show that the up-regulation of synaptic AMPARs induced by deprivation is dependent on expression of the AMPAR-binding protein GRIP1. Collectively, our study reveals detailed spatiotemporal dynamics of AMPARs within live animals in response to visual deprivation.

## Results

### Long-term detection of spine SEP-GluA1 in V1 L2/3 neurons in vivo

To track AMPAR and spine dynamics in L2/3 neurons of V1 in awake mice, we employed in utero electroporation to transfect L2/3 pyramidal neurons with the GluA1 AMPAR subunit tagged with Super Ecliptic pHluorin (SEP), a pH-sensitive form of green fluorescent protein, myc-GluA2 AMPAR subunit, and dsRed2 as previously described (*Makino and Malinow, 2011*; *Suresh and Dunaevsky, 2017*; *Zhang et al., 2015*; *Figure 1*; *Figure 1—figure supplement 1*). First, we found a high correlation between spine intensity of SEP-GluA1 and glutamate uncaging-evoked excitatory postsynaptic current (uEPSC) amplitude (*Figure 1C,D*), suggesting that spine enrichment of SEP-GluA1 largely reflects postsynaptic strength. Next, we repeatedly imaged the same dendritic spines in adult mice (P70 - P85) over 10 days (*Figure 1A, B, E*). To control for day-to-day variability in imaging conditions such as laser intensity and window quality, we normalized SEP-GluA1 spine and dendrite intensity and spine dsRed intensity to dendritic shaft dsRed2 intensity which would not be expected to change (*Zhang et al., 2015*). We found that the majority of spines in V1 L2/3 neurons (68%) persisted throughout all 10 imaging days (*Figure 1F*), comparable to previous reports (*Holtmaat et al., 2005*), suggesting that the modest overexpression of AMPARs does not affect spine dynamics. Total spine surface GluA1 (sGluA1) levels on a dendrite, average spine sGluA1 expression and spine size in persistent spines were all stable (*Figure 1G–I*). These results show that in adult animals, overall expression of spine sGluA1 expression remains constant throughout daily imaging sessions.

### Dynamic baseline expression of sGluA1 within individual spines

Since we were able to monitor individual synapse dynamics over time with two-photon imaging, we next examined expression of sGluA1 within individual spines. We observed that sGluA1 intensity within individual persistent spines was highly dynamic and varied over days (*Figure 2A*). Despite the dynamic expression across days, the relative sGluA1 level in individual spines at day 1 remained strongly correlated with their level at day 10 (*Figure 2B*), indicating that the difference in sGluA1 level between persistent spines doesn't change over time with strong spines remaining strong and weak spines staying weak.

Next we measured the dynamics of spine sGluA1 expression across days by calculating the coefficient of variance (CV) of sGluA1 signal from spines and investigated the underlying mechanisms. Intriguingly, we did not observe any correlations between CV of spine sGluA1 signal and their initial spine sGluA1 level (*Figure 2C*), suggesting that all spines show similar extent of dynamics despite their variances in sGluA1 levels. Structural spine dynamics including spine formation and elimination have been reported and it has been shown that the structural dynamics are dependent on sensory input (*Holtmaat and Svoboda, 2009*; *Majewska and Sur, 2003*). To assess whether the functional dynamic of AMPAR expression observed here is also dependent on experience or sensory input, we visually deprived mice using binocular enucleation and measured the ensuing dynamics. We found

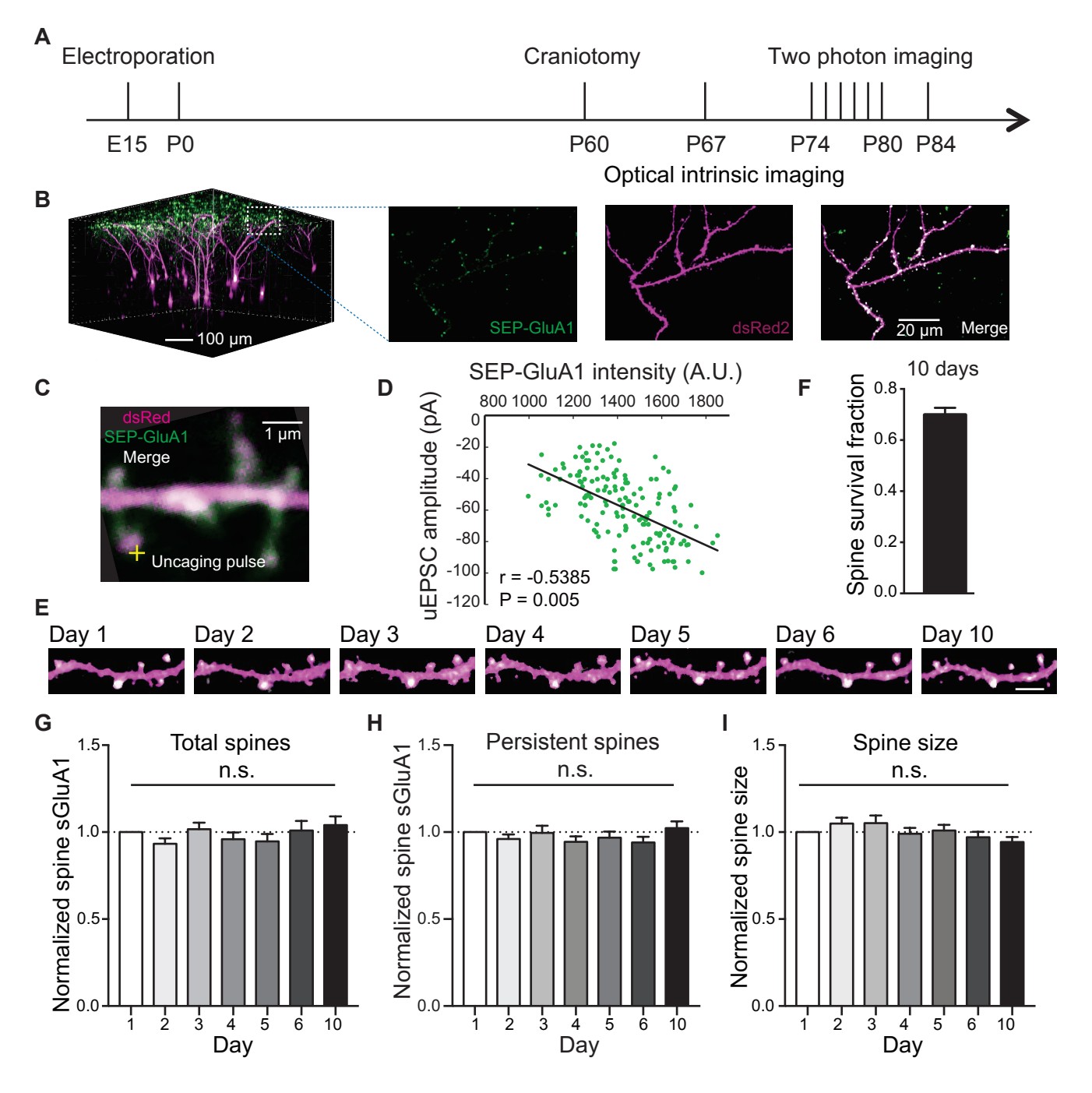

**Figure 1.** Long-term detection of spine SEP-GluA1 in V1 L2/3 neurons in vivo. (A) Experimental timeline. (B) (Left) 3D reconstruction of L2/3 neurons of visual cortex transfected with SEP-GluA1 (green) and dsRed2 (magenta), merged in white. (Right) Z projection of imaging volume in white box. (C) Example of a whole-cell recording and glutamate uncaging at a spine with high sGluA1 enrichment. (D) Correlation between spine sGluA1 intensity and uEPSC amplitude (n = 146 spines; Pearson). (E) Representative time-lapse images of V1 L2/3 apical dendrites. Scale bar: 5 $\mu m$. (F) Percentage of spines that persist across 10 imaging days. (G) Total spine sGluA1 level on the dendrite did not change significantly with time. Total spine sGluA1 level on a dendrite was calculated by summing all spines on the same dendrite (n = 19 dendrites from five mice; one-way ANOVA). (H) Stable expression of average spine sGluA1 in persistent spines over 10 days (n = 23 dendrites from six mice; one-way ANOVA). (I) Average spine size in persistent spines had no significant change over days (n = 23 dendrites from six mice; one-way ANOVA).

The online version of this article includes the following figure supplement(s) for figure 1:

**Figure supplement 1.** Expression of SEP-GluA1 in L2/3 neurons.

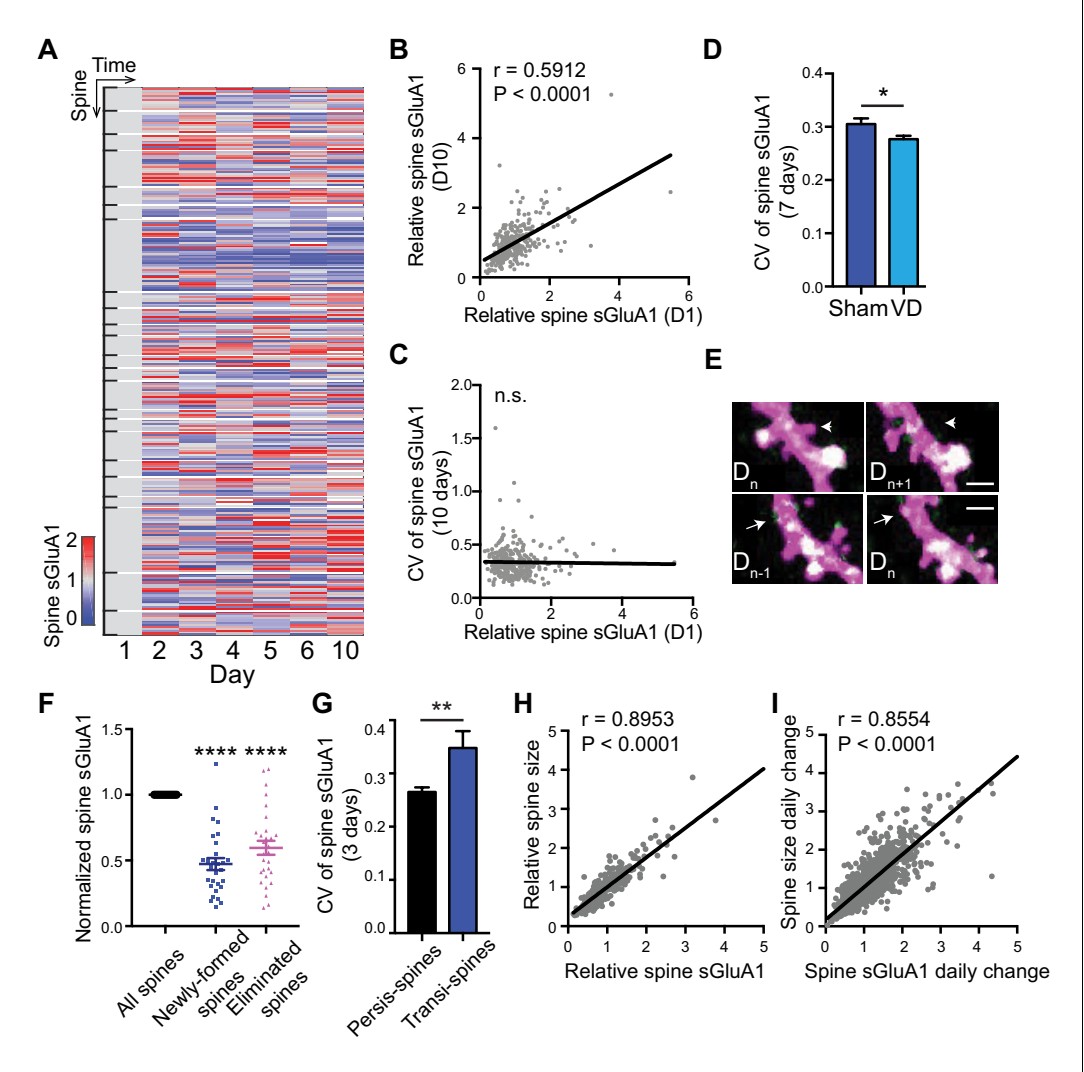

**Figure 2.** Dynamic baseline expression of sGluA1 within individual spines. (A) Heat map of sGluA1 expression within individual spines. Spines on the same dendrites were grouped together. (B) Correlation of spine sGluA1 level between day 1 and day 10 in V1 L2/3 neurons (n = 279 spines; Pearson). (C) Correlation between spine sGluA1 level at day 1 and its CV in V1 L2/3 neurons (n = 280 spines; Pearson). (D) The CV of spine sGluA1 signal in V1 L2/3 neurons before and after visual deprivation (VD) (n = 280/520 (Sham/VD) spines in V1 L2/3 neurons; Student's t-test). (E) Representative images of newly-formed spines (bottom) and eliminated spines (upper). Scale bars: 2 μm. (F) sGluA1 levels in newly-formed spines and eliminated spines (n = 29 spines; Student's t-test). (G) The CV of spine sGluA1 expression in persistent and transient spines in V1 L2/3 neurons (n = 327 persistent spines; n = 34 transient spines). (H) Correlation between spine size and spine sGluA1 intensity on imaging day 1 (n = 280 spines; Pearson). (I) Correlation between daily change in spine size and daily change in spine sGluA1 level (n = 1384 spines; Pearson). Data are presented as mean ± SEM. n.s., not significant; *p<0.05; **p<0.01; ****p<0.0001.

The online version of this article includes the following figure supplement(s) for figure 2:

**Figure supplement 1.** Dynamic sGluA1 expression within individual spines.

that the CV of V1 L2/3 neurons became significantly smaller after deprivation (*Figure 2D*). This demonstrates that AMPAR dynamics in individual spines is partially driven by sensory input.

We also characterized sGluA1 expression in non-persistent (transient) spines that were newly formed or eliminated, and observed that both groups of spines had significantly lower sGluA1 expression compared with persistent spines on the same dendrite (*Figure 2E,F*). Nevertheless, newly formed spines gradually increased sGluA1 levels after formation, while eliminated spines significantly

decreased sGluA1 content prior to elimination (*Figure 2—figure supplement 1A,B*). The CV of spine sGluA1 expression in transient spines was much higher than that in persistent spines (*Figure 2G*), suggesting that GluA1 levels on transient spines are more dynamic than on persistent spines. These results indicate that sGluA1 level largely reflects spine stability.

It is widely believed that the number of AMPARs in spines is strongly correlated with spine size (*Nusser et al., 1998*; *Takumi et al., 1999*). To test this, we quantified AMPAR expression relative to spine size as indicated by dsRed2 soluble-cell fill intensity. Similar to spine sGluA1 expression, persistent spines did not significantly vary in their average size across all time points (*Figure 1I*), although individual spines did display size fluctuation over the measured time course (*Figure 2—figure supplement 1C*). Similar to spine sGluA1 level, the relative spine size in individual spines at day 1 was strongly correlated with their size at day 10 (*Figure 2—figure supplement 1D*). Consistent with previous in vitro studies, we found a strong positive correlation between spine size and spine sGluA1 intensity in vivo (*Figure 2H*). Moreover, there was a highly positive correlation between the daily change of spine size and sGluA1 change (*Figure 2I*), indicating that spine size and GluA1 expression increase or decrease concurrently. However, we identified a small fraction of spines (~18%) that show an inverse correlation between daily spine size change and sGluA1 change (*Figure 2—figure supplement 1E,F*). We further examined the sGluA1 level distribution of those spines and found that there was a significant leftward shift of sGluA1 level distribution in those subsets of spines compared to all spines (*Figure 2—figure supplement 1G*), suggesting that they tend to be spines with lower sGluA1 level.

## V1-specific spine sGluA1 increases following binocular deprivation

Visual deprivation via dark exposure has been shown to induce homeostatic plasticity in V1 L2/3 neurons using ex-vivo measurements, wherein AMPARs in synaptosome preparations are increased following deprivation (*Goel et al., 2006*; *Goel and Lee, 2007*). To examine in vivo changes in the same V1 neurons, we repeatedly imaged apical dendrites from L2/3 neurons of awake mice before and after visual deprivation (VD) by binocular enucleation. Enucleation did not affect dendritic dsRed2 signal nor spine survival rate (*Figure 3—figure supplement 1A,B*). However, the average synaptic sGluA1 level in persistent spines of V1 neurons decreased 1 day after enucleation, recovered at day 2, and significantly increased by day 7 (*Figure 3A,B*; *Figure 3—figure supplement 1C*). The changes of spine size showed similar trends but with smaller changes (*Figure 3C*). We also imaged spines in visual cortical regions outside of V1 (non-V1) and found that there were no significant increases of sGluA1 expression, as well as spine size, in non-V1 visual cortex following 7 days of enucleation (*Figure 3D–F*). Indeed, we observed a systematic decrease in sGluA1 level after VD in those regions (*Figure 3D,E*). These data demonstrate that visual deprivation-induced spine enrichment of sGluA1 is specific to V1.

## Heterogeneous responses of individual spines to visual deprivation

As our imaging approach enables us to track changes of individual spines over time, we next examined how individual dendrites or spines of V1 L2/3 neurons responded to visual deprivation. We found that the responses of individual dendrites and spines were highly heterogeneous (*Figure 4A*; *Figure 4—figure supplement 1A*). Despite the overall increase in spine sGluA1 expression after 7 days of deprivation, some dendrites or spines did reduce sGluA1 expression (*Figure 4A*; *Figure 4—figure supplement 1A*), indicating that only a subset of spines undergo potentiation. However, compared to control mice with sham-surgery (SH), a greater proportion of dendrites or spines underwent increases (*Figure 4A, B*; *Figure 4—figure supplement 1B*). In the heatmap, we noticed that dendrites that showed decreases at day 7 tended to decrease sGluA1 level at day 1 following deprivation while the dendrites exhibiting increases of sGluA1 expression at day 7 did not show obvious decreases (*Figure 4A*). Therefore, we examined the relationship between spine sGluA1 changes of individual dendrites at day 1 and at day 7 after enucleation. A very strong positive correlation was detected (*Figure 4C*), indicating that dendrites that display a rapid decrease at day 1 would remain decreased at day 7 or show a reduced increase by day 7. We then separately plotted the dendrites that did or did not display a decrease after 1 day of enucleation and examined the relative sGluA1 levels of each group at day 7 (for details, see Methods). We observed two populations of dendrites: one population showed decrease of sGluA1 at day 1 but recovered to baseline level by day 7; The

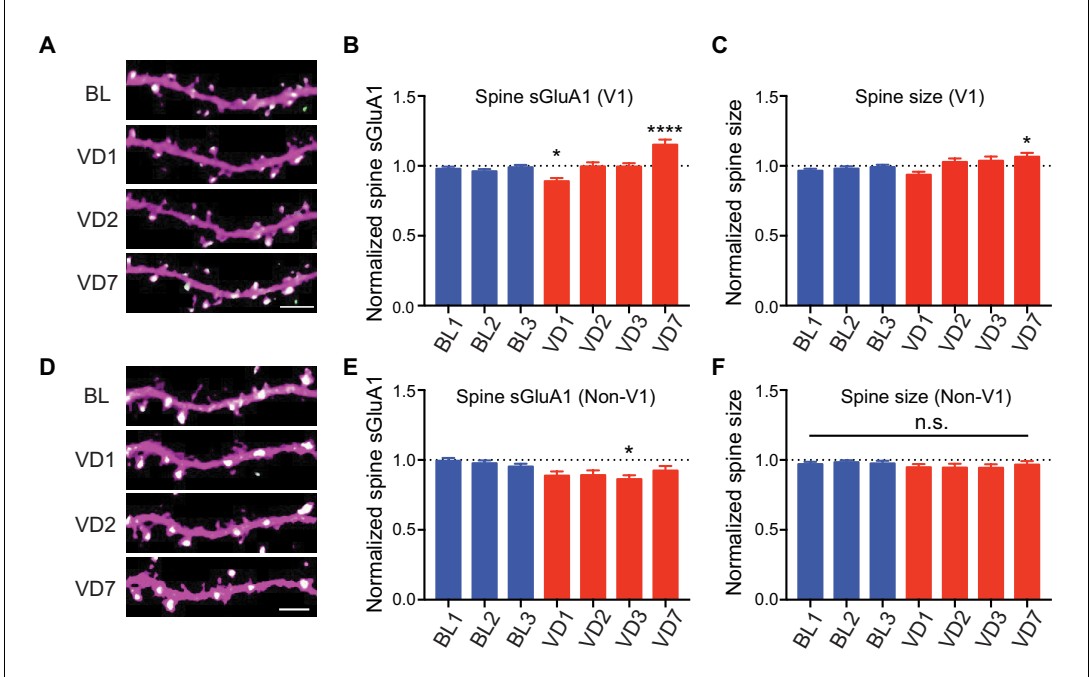

**Figure 3.** V1-specific spine sGluA1 increases following binocular deprivation. (A–C) Changes in spine sGluA1 expression and spine size before (baseline, BL) and after visual deprivation (VD) in V1 L2/3 neurons (n = 49 dendrites from eight mice; one-way ANOVA). Scale bar: 5 µm. (D–F) Changes in spine sGluA1 expression and spine size following VD in non-V1 visual cortex (n = 33 dendrites from six mice; one-way ANOVA). Scale bar: 5 µm. Data are presented as mean ± SEM. n.s., not significant; *p<0.05; ****p<0.0001.

The online version of this article includes the following figure supplement(s) for figure 3:

**Figure supplement 1.** V1-specific spine sGluA1 increases following binocular deprivation.

other population of dendrites did not change at day 1 but showed significant up-regulations of sGluA1 at day 7 (*Figure 4D*). The percentage of decrease dendrites was about 51% (*Figure 4E*, for details, see Methods). These two distinct populations of dendrites did not result from different populations of neurons, as we found that dendrites from the same neuron still responded differently to deprivation (*Figure 4F*). Similarly, we observed two populations of spines with distinct responses to the enucleation (*Figure 4—figure supplement 1C*). Together, these results show that the responses of individual spines or dendrites to visual deprivation are highly heterogeneous.

## Depth-dependent changes in spine sGluA1 expression after visual deprivation

Previous studies using electrophysiological recordings in acute brain slices have reported that 2 days of visual deprivation is sufficient to increase AMPAR-mediated mEPSC amplitude in V1 L2/3 neurons (*Bridi et al., 2018*; *Goel and Lee, 2007*), however, we didn't observe any changes in synaptic AMPAR expression at that time point in vivo. To address this, we performed biochemical experiments by dissecting V1 and isolating synaptosomes to examine synaptic AMPAR levels from mice 2 days after binocular enucleation or sham-surgery. In agreement with previous studies (*Bridi et al., 2018*; *Goel and Lee, 2007*), we observed a significant increase in synaptic GluA1 but not GluN1 in enucleated mice (*Figure 5A,B*). We next asked if the discrepancy in the imaging and biochemical results could result from analyzing distinct dendritic compartments, as in vivo imaging biases examination of more dorsal L1 synapses whereas in vitro biochemical experiments sample L1-L6 (*Figure 5—figure supplement 1A*). We isolated L1 and L2-6 by micro-dissecting V1 cortical tissue and then examined synaptic AMPAR expression separately (*Figure 5—figure supplement 1A*). The experiment revealed that synaptic GluA1 in L1 was decreased 2 days after enucleation but subsequently recovered to baseline by day 7, whereas synaptic GluA1 expression in L2-6 significantly increased following 2 days of enucleation and remained enhanced at day 7 (*Figure 5C,D*). These

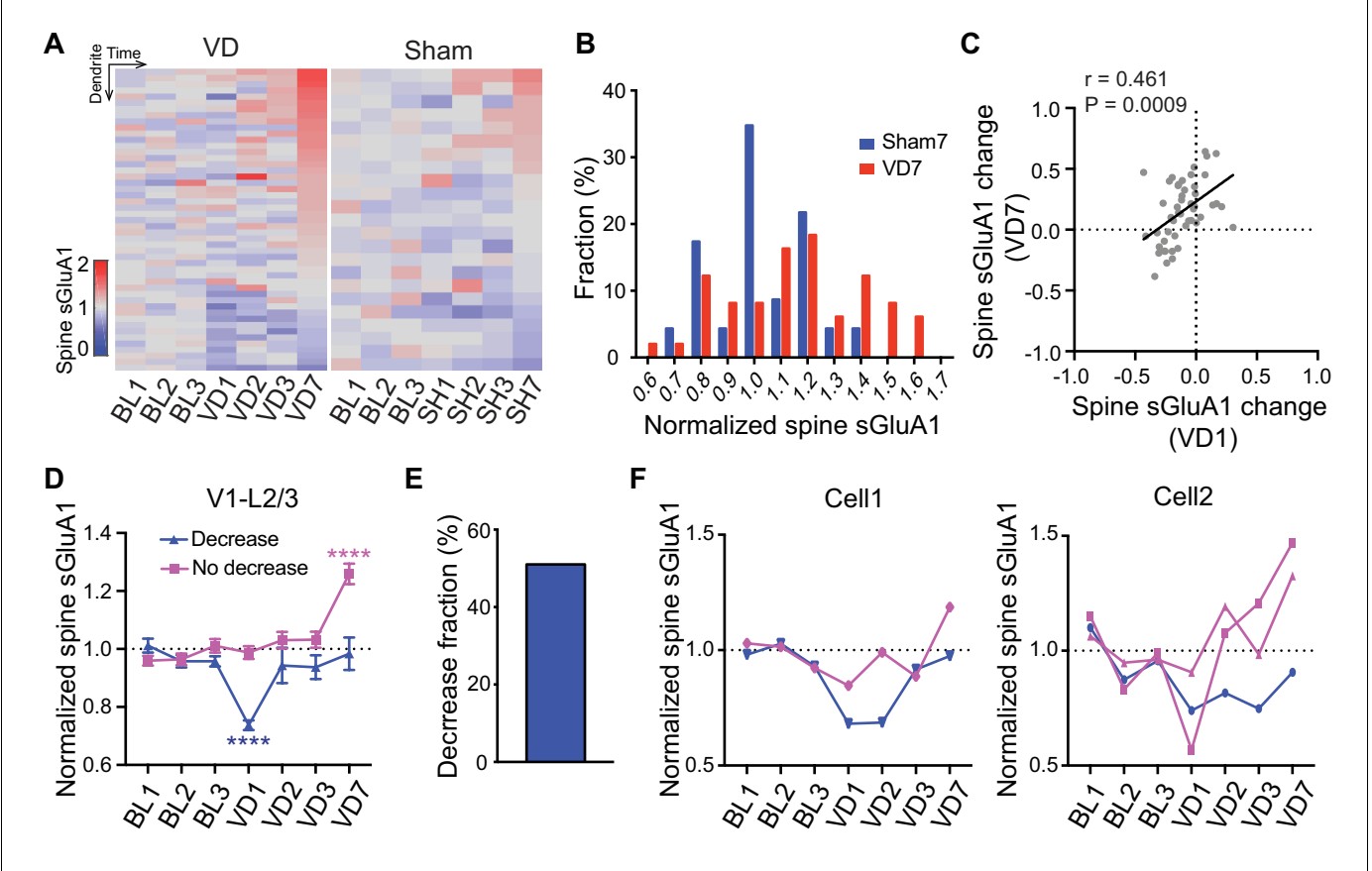

**Figure 4.** Heterogeneous responses of individual spines to visual deprivation. (A) Heat map of change in sGluA1 level within individual dendrites in VD and sham-surgery (SH) groups. (B) Histogram of spine sGluA1 changes in dendrites 7 days after VD or sham-surgery (n = 23 dendrites in sham group; n = 49 dendrites in VD group; Kolmogorov-Smirnov test, p=0.079). (C) Correlation between spine sGluA1 changes of individual dendrites at day 1 and day 7 after VD (n = 49 dendrites; Pearson). (D) Changes of spine sGluA1 expression in dendrites that show decrease or no decrease at day 1 following VD in V1 L2/3 neurons. (n = 25 decrease dendrites and n = 24 no decrease dendrites from eight mice; one-way ANOVA). (E) Percentage of dendrites that decrease spine sGluA1 after 1 day of VD (51.02%). (F) Changes in spine sGluA1 of individual dendritic segments from the same V1 L2/3 neuron following binocular enucleation. Each line indicates individual dendritic segment. Data are presented as mean ± SEM. ****p<0.0001.
The online version of this article includes the following figure supplement(s) for figure 4:

**Figure supplement 1.** Heterogeneous responses of individual spines to visual deprivation.

data suggest that visual deprivation has distinct effects on AMPAR expression in spines located in different layers. To verify this in vivo, we repeatedly imaged basal dendrites from L2/3 neurons (150–300 μm deep) before and after enucleation. The amount of spine sGluA1 on basal dendrites was significantly increased 1 day following enucleation and remained elevated afterwards (*Figure 5E,F*; *Figure 5—figure supplement 1B,C*). To confirm that the difference in time course between basal dendrites and apical dendrites in response to deprivation is dependent on locations rather than resulting from different populations of neurons being examined, we imaged both apical dendrites and basal dendrites from the same neurons. We found that spine sGluA1 increases were consistently and significantly larger on basal dendrites than on apical dendrites (*Figure 5G*). The ratios of relative change in spine sGluA1 from basal dendrites over apical dendrites were significantly larger than one following deprivation (*Figure 5H*). Taken together, these data reveal that spines on basal dendrites of L2/3 neurons increase GluA1 content faster and more robustly than spines on apical dendrites in response to visual deprivation.

To further investigate depth-dependent mechanisms of sGluA1 expression, we examined the relationship between the depth of dendrites from the pia mater and changes of sGluA1 in dendrites. The change of average spine sGluA1 expression level on dendrites after 7 days of enucleation was

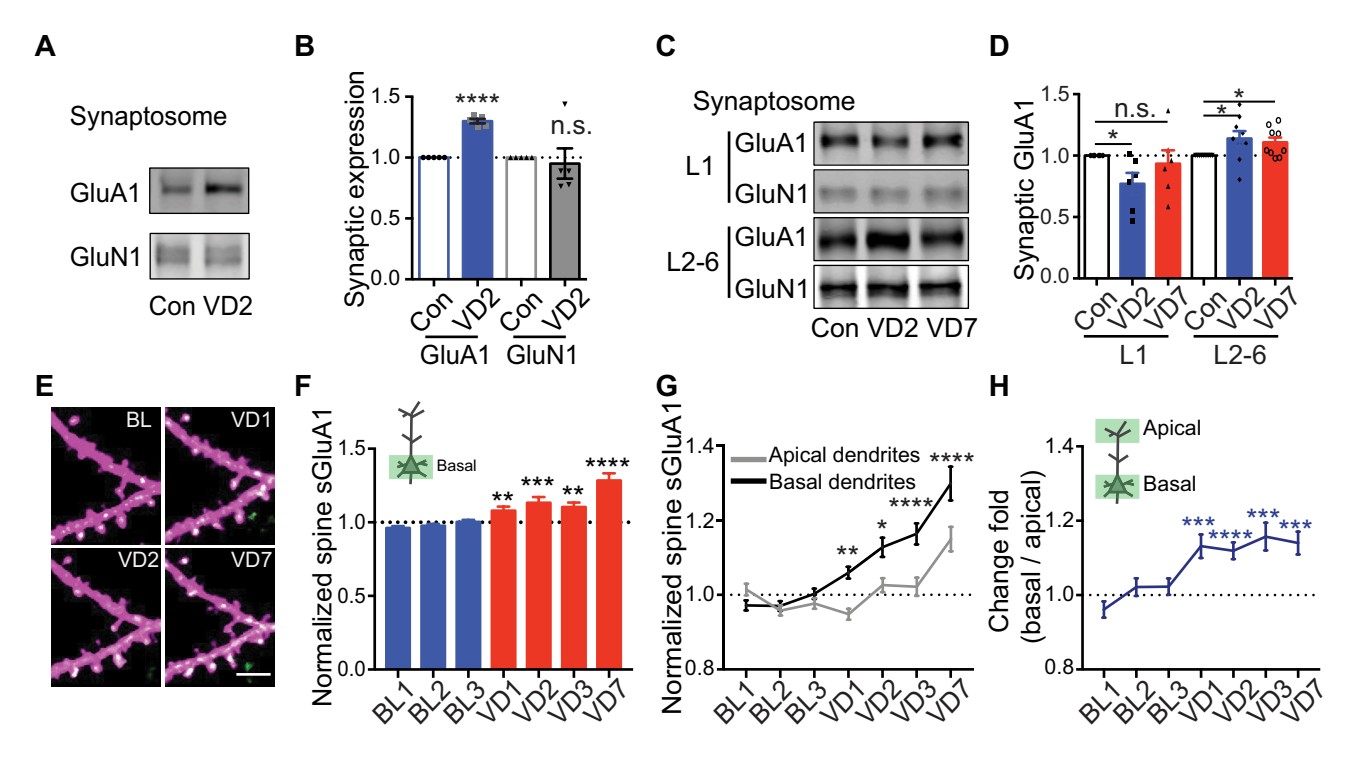

**Figure 5.** Depth-dependent changes in spine sGluA1 expression after visual deprivation. (A and B) Synaptic GluA1 and GluN1 levels in V1 from 2 days' enucleated or sham-surgery mice (n = 5; Student's t-test). (C and D) Synaptic GluA1 levels from superficial (L1) and deep (L2-6) layers of V1 (n = 6–9; Student's t-test). (E and F) Changes in spine sGluA1 on basal dendrites of V1 L2/3 neurons following VD (n = 40 dendrites from six mice; one-way ANOVA). Scale bar: 5 μm. (G) Changes in spine sGluA1 on basal and apical dendrites from the same neurons following VD (n = 16 neurons from seven mice; repeated measure two-way ANOVA). (H) Change ratios of basal dendrites to apical dendrites of the same neurons following VD were significantly larger than 1 (n = 16 neurons from seven mice. one sample t-test). Data are presented as mean ± SEM. n.s., not significant; *p<0.05; **p<0.01; ***p<0.001; ****p<0.0001.

The online version of this article includes the following figure supplement(s) for figure 5:

**Figure supplement 1.** Depth-dependent increase of spine sGluA1 in V1-L2/3 neurons.

positively correlated with the depth of dendrites (*Figure 5—figure supplement 1D,E*), indicating that deep dendrites have greater increases than superficial dendrites. Next, we tested whether spines along the same dendrite had similar depth-dependent changes. To distinguish between the depth of spines and the distance from dendrite branch point, we separately analyzed ascending dendrites and descending dendrites. In both ascending dendrites and descending dendrites, there was a positive correlation between spine depth and deprivation-induced sGluA1 expression (*Figure 5—figure supplement 1F*), such that deep spines increase sGluA1 more than superficial spines following deprivation. Significantly, the correlation of spine sGluA1 change with the distance from dendrite branch point showed opposite directions in ascending and descending dendrites (*Figure 5—figure supplement 1G*), suggesting that changes in sGluA1 expression are correlated with spine depth instead of distance from dendrite branch point. This depth-dependent change in spine sGluA1 was not caused from imaging artifacts. First, such correlation was not observed in sham-surgery mice (*Figure 5—figure supplement 1H*). Second, when we focused on baseline expression of spine sGluA1, we found it was positively correlated with the distance from branch point but not with the depth of spines (*Figure 5—figure supplement 1I,J*). Because these correlations were weak but statistically significant, we further analyzed synaptic sGluA1 signal intensity in apical and basal dendritic compartments within each imaging session to determine if we would observe similar depth-dependent effects on sGluA1 expression following enucleation. We categorized spines by dendritic compartment and then compared the sGluA1 intensity in the deepest 15% of spines with the most superficial 15% of spines along the same dendritic segment. In both apical and basal dendrites we

found that spines positioned more deeply from the pia exhibited greater deprivation-induced changes in synaptic sGluA1 expression than spines more superficially poised along the same dendrite (*Figure 5—figure supplement 1K*). Collectively, these data demonstrate that visual deprivation induces depth-dependent changes in spine sGluA1.

## Lamina-specific increases of spine sGluA1 level after deprivation

Laminar differences have been reported for ex vivo measurements of homeostatic synaptic scaling induced by visual deprivation, where L4 scaling occurs only before P21, and L2/3 scaling is observed only after this age (*Desai et al., 2002*; *Petrus et al., 2011*). However, whether L5 neurons are involved in scaling remains largely unknown. Additionally, 7 days of enucleation did not induce a change of synaptic GluA1 expression in L1 in our in vitro biochemical experiments but caused a significant increase in spine sGluA1 level on apical dendrites of L2/3 neurons in vivo. As L5 neurons also have dendrites in L1, we asked if L5 neurons responded to visual deprivation. We similarly used in utero electroporation to transfect V1 L5 neurons and longitudinally imaged apical dendrites exclusively from these neurons (*Figure 6A,B*). Under baseline conditions we found that average spine sGluA1 expression and spine size on apical dendrites of L5 neurons were very stable with 66% spines persisting across 10 imaging days (*Figure 6—figure supplement 1A–D*). Similar to L2/3 neurons, sGluA1 intensity within individual spines on apical dendrites of L5 neurons was very dynamic as well. However, the average CV of spine sGluA1 signal on apical dendrites of L5 neurons was significantly smaller than that on apical dendrites of L2/3 neurons (*Figure 6C*), indicating that L5 neurons are less dynamic than L2/3 neurons. Following visual deprivation, the spine survival rate of L5 neurons remained unchanged (*Figure 6—figure supplement 1D*), but the CV of spine sGluA1 level was increased (*Figure 6D*), further suggesting that this dynamic is dependent on sensory input. Regarding synaptic AMPAR expression, we observed that spine sGluA1 expression was reduced after 1 day and then recovered to baseline levels (*Figure 6E,F*; *Figure 6—figure supplement 1E*). Nevertheless, in contrast to L2/3 neurons, we didn't observe any significant increase after 7 days of deprivation in L5 neurons (*Figure 6E,F*; *Figure 6—figure supplement 1E,F*). We therefore conclude that visual deprivation specifically enhances AMPAR expression on apical dendrites of V1 L2/3 neurons but not on apical dendrites of L5 neurons, intriguingly suggesting exclusive mechanisms that regulate these distinct cortical circuits.

We next examined how individual dendrites or spines in L5 responded to visual deprivation. Again, we found that the responses were very heterogeneous, with some dendrites increasing sGluA1 and some decreasing sGluA1 (*Figure 6—figure supplement 1G*). We also observed a similar positive correlation of spine sGluA1 changes between day 1 and day 7 following visual deprivation in L5 neurons (*Figure 6G*). We then categorized cells based on whether their apical dendrites did or did not display a decrease after 1 day of enucleation, and we found that there were two populations of cells with distinct responses to binocular deprivations: one group showed decrease at day 1 and day 2 after enucleation and then recovered to baseline later while the other group did not decrease at day 1 but exhibited a gradual increase following enucleation despite that increases were not significant at any time points compared to the baseline level (*Figure 6H*). Previous studies in the barrel cortex indicate that specific subtypes of L5 neurons respond differentially to changes in sensory experiences (*Greenhill et al., 2015*; *Holtmaat et al., 2006*). The two populations of cells in V1 L5 with distinct responses to visual deprivation observed here might be due to their different cell types.

We also investigated whether visual deprivation induced a similar depth-dependent change of spine sGluA1 in L5 neurons as in L2/3 neurons despite the observation that there is no net increase. To accomplish this, we analyzed only ascending L5 dendrites since there are few descending apical L5 dendrites in L1. The change of synaptic sGluA1 expression on dendrites did not correlate with the depth (*Figure 6—figure supplement 1H*). For individual spines, in both sham-surgery and VD groups, we did not see any correlations between the change in spine sGluA1 and spine depth (*Figure 6—figure supplement 1I*). Nevertheless, there was a strong positive correlation between baseline spine sGluA1 level and the distance to dendrite branch point in L5 neurons (*Figure 6—figure supplement 1J*). These results indicate that the depth-dependent changes of spine sGluA1 induced by deprivation are specific to L2/3 neurons as well. However, the distance-dependent baseline expression of spine sGluA1 occurs in both L2/3 and L5 neurons, suggesting that this is a more general phenomenon.

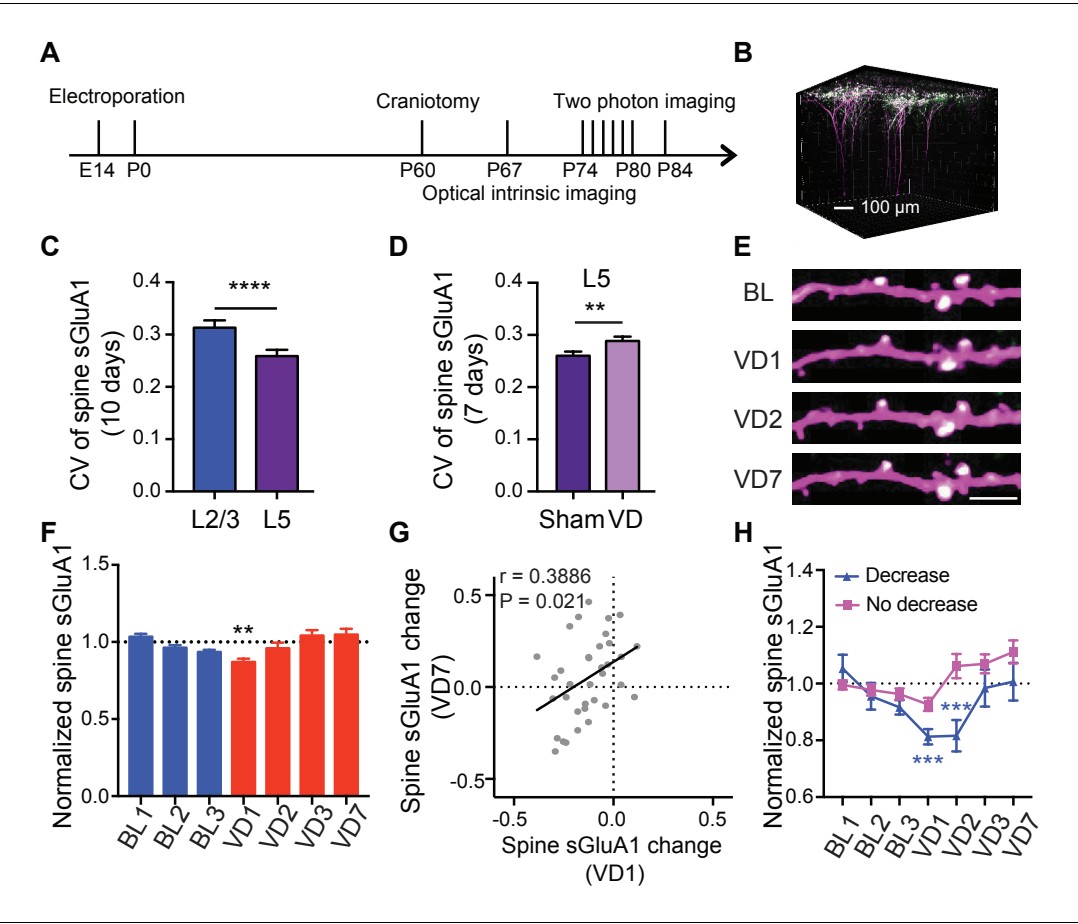

**Figure 6.** Lamina-specific increases of spine sGluA1 following visual deprivation. (**A**) Experimental timeline. (**B**) 3D reconstruction of L5 neurons of visual cortex transfected with sGluA1 (green) and dsRed2 (magenta), merged in white. (**C**) Average CV of spine sGluA1 expression in V1 L2/3 and L5 neurons under baseline conditions. (n = 280 spines, L2/3 neurons; n = 256 spines, L5 neurons; Student's t-test). (**D**) The CV of spine sGluA1 in L5 neurons before and after visual deprivation. (n = 256/277 (Sham/VD) spines in L5 neurons; Student's t-test). (**E**) Representative time-lapse images of L5 apical dendrites. Scale bar: 5 μm. (**F**) Changes in spine sGluA1 expression on apical dendrites of V1 L5 neurons following VD (n = 35 dendrites from four mice; one-way ANOVA). (**G**) Correlation between spine sGluA1 changes of individual dendrites in L5 neurons at day 1 and day 7 after VD (n = 35 dendrites; Pearson). (**H**) Changes of spine sGluA1 expression in cells that show decrease or no decrease at day 1 following VD in L5 neurons. (n = 6 decrease cells and n = 6 no decrease cells from four mice; one-way ANOVA). Data are presented as mean ± SEM. **p<0.01; ***p<0.001; ****p<0.0001.

The online version of this article includes the following figure supplement(s) for figure 6:

**Figure supplement 1.** Lamina-specific increases of spine sGluA1 following visual deprivation.

## GRIP1-dependent increases of sGluA1 following deprivation

Finally, we sought to determine the mechanisms underlying the increase of synaptic AMPARs following visual deprivation in vivo. Glutamate receptor interacting protein 1 (GRIP1) is a multi-PDZ domain containing protein that binds directly with AMPAR subunits (*Dong et al., 1997*). We have previously shown that GRIP1 plays a key role in regulating AMPAR trafficking, synaptic targeting and homeostatic plasticity (*Gainey et al., 2015*; *Mao et al., 2010*; *Pfennig et al., 2017*; *Tan et al., 2015*). To test whether GRIP1 mediates the increase in synaptic GluA1 induced by visual deprivation, we generated *Grip1* conditional knockout mice (neuron-specific deletion via Nestin-Cre expression) (*Mejias et al., 2011*). In wild type (WT) mice, using biochemical experiments we identified a significant increase in synaptic GluA1 as well as GRIP1 in V1 after 2 days of enucleation as described above (*Figure 7A,B*). However, no increase in synaptic GluA1 was observed in *Grip1* knockout mice

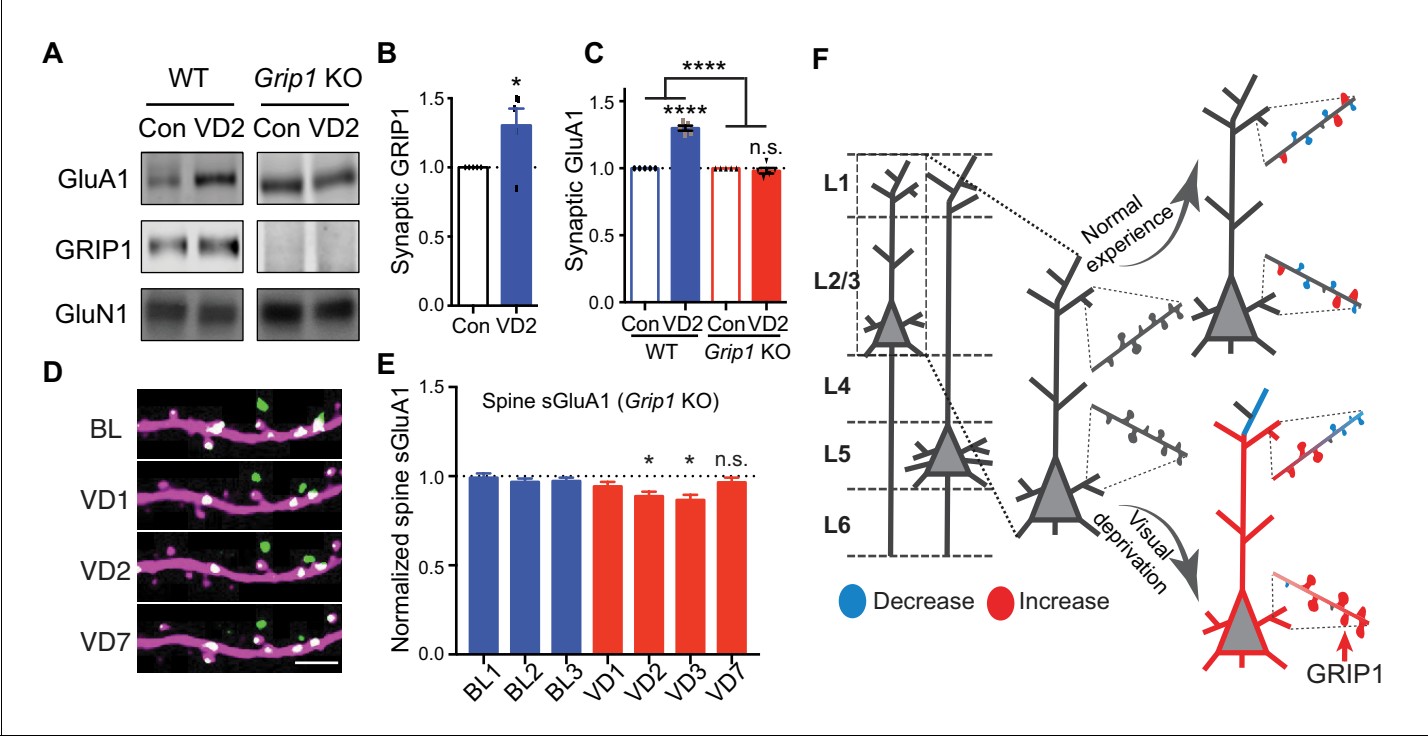

**Figure 7.** GRIP1-dependent increases of sGluA1 following deprivation. (**A–C**) Synaptic GluA1 and GRIP1 levels in V1 from WT and Grip1 knockout (KO) mice with 2 days of sham-surgery or VD (n = 5; Student's t-test and two-way ANOVA). (**D–E**) Changes in spine sGluA1 expression on apical dendrites of V1 L2/3 neurons from Grip1 KO mice following VD (n = 32 dendrites from five mice; one-way ANOVA). Scale bar: 5 μm. (**F**) Model of spine GluA1 dynamics in mice with normal experience or visual deprivation. Data are presented as mean ± SEM. n.s., not significant; *p<0.05; ****p<0.0001. The online version of this article includes the following figure supplement(s) for figure 7:

**Figure supplement 1.** GRIP1-dependent changes in V1 L2/3 neurons following visual deprivation.

(*Figure 7A,C*). Moreover, we observed no increases of spine sGluA1 on apical dendrites of V1 L2/3 neurons in *Grip1* knockout mice in vivo following enucleation. In fact, we observed a small decrease in spine sGluA1 level that recovered between day 3 and day 7 after deprivation (*Figure 7D,E*; *Figure 7—figure supplement 1A,B*). The initial decrease of sGluA1 after enucleation phenocopies our results from WT mice (*Figure 3B*) and might be extended in *Grip1* knockout mice due to reduced AMPAR exocytosis. The later recovery could be some compensatory regulations by other AMPAR-binding proteins, like GRIP2 (*Anggono and Huganir, 2012*). Nevertheless, these data demonstrate that GRIP1 is essential for the deprivation-induced up-regulation of synaptic AMPARs.

## Discussion

In the present study, we chronically monitored AMPAR expression in individual synapses within live animals with or without binocular enucleation in order to investigate how experience shapes neural circuits in the adult brain. We found that under baseline conditions in mice with normal experience, sGluA1 expression levels in individual spines are very dynamic. Upon visual deprivation, basal dendrites of V1 L2/3 neurons enhance sGluA1 earlier than apical dendrites and deep spines increase more than superficial spines. The changes induced by deprivation are specific to V1 L2/3 neurons but not L5 neurons and the increase in spine sGluA1 is dependent on GRIP1 expression (*Figure 7F*). To our knowledge, this work is the first to longitudinally examine synapse strength in different layers of cortical neurons in unanesthetized awake mice. Our in vivo imaging data with single-synapse resolution provide unprecedented levels of spatiotemporal information regarding synaptic AMPAR dynamics and reveal that neurons exhibit a tremendous heterogeneity of synaptic changes under both normal and sensory deprivation conditions.

AMPAR trafficking is critical for synaptic plasticity and brain function (*Anggono and Huganir, 2012*; *Huganir and Nicoll, 2013*; *Volk et al., 2015*). We show that synaptic expression of AMPARs is highly dynamic in mice with normal experience despite the relatively stable overall expression, suggesting that strength of individual synapses is continuously being modified. Intriguingly, the degree of variation of spine sGluA1 expression in L2/3 neurons is larger than that in L5 neurons, indicating that spines in L2/3 neurons are more dynamic than those in L5 neurons. Since L2/3 neurons receive more feedforward inputs from L4 than L5 neurons, the higher dynamic observed in L2/3 neurons suggests that sensory input is a drive for synaptic AMPAR dynamic. Moreover, depriving the visual input greatly changes the dynamics of V1, shifting from L2/3 to L5, which further supports that this dynamic is driven by sensory inputs.

Synaptic strength and spine size are well correlated, and synaptic potentiation or depression is usually accompanied with an increase or decrease in spine size, respectively (*Bosch et al., 2014*; *Makino and Malinow, 2009*). In agreement with previous studies, overall we observed a strong correlation between spine sGluA1 level and spine size in live animals. Further, we found that the change of spine sGluA1 and change in spine size change were also highly correlated. Nevertheless, we do find a small population of spines that exhibit a divergence of spine form and function. Moreover, synaptic sGluA1 shows greater changes than spine size following visual deprivation. These suggest that differences in synaptic strength may be underestimated or even not correctly determined when using spine size as a measurement. Indeed, the dissociation of spine size and synaptic strength has been reported many times (*Lee et al., 2012*). Spine number or volume is not changed at all at cerebellar Purkinje cell synapses during LTD (*Sdrulla and Linden, 2007*). Insulin-induced endocytosis of AMPARs is not accompanied by spine shrinkage (*Wang et al., 2007*). Spine size and AMPAR function are coupled through some common signaling mechanisms, but there is a divergence in the downstream signaling pathways that regulate these two processes. For instance, it has been shown that spine shrinkage is mediated by cofilin while LTD is dependent on protein phosphatase one although both require calcium influx through NMDA receptors and enhanced calcineurin activity (*Zhou et al., 2004*). Thus, spine size, in certain conditions, is not a good indication of synaptic strength, and our imaging of synaptic AMPAR expression provides a direct and accurate way to monitor functional changes at synapses.

Neurons receive and process information from thousands of inputs at synapses that are distributed throughout the extensively branching dendrites. To overcome the filtering and attenuation caused by the cable properties of dendrites (*Rall, 1962a*; *Rall, 1962b*), synapses express a varying number of AMPARs that increases with distance from the soma, a phenomenon known as distance-dependent scaling (*Andrasfalvy and Magee, 2001*). While this phenomenon has been extensively studied in hippocampal CA1 pyramidal neurons (*Menon et al., 2013*; *Nicholson et al., 2006*; *Shipman et al., 2013*), it is unknown whether this scaling occurs in cortical pyramidal neurons. Here, we find that both L2/3 and L5 cortical pyramidal neurons display a distance-dependent baseline expression of sGluA1, wherein distal spines have higher levels of sGluA1 than proximal spines. These results suggest that the distance-dependent scaling of AMPARs might be a general phenomenon across brain regions. The molecular mechanisms of this phenomenon and whether the underlying mechanisms are same or different between hippocampal and cortical pyramidal neurons require further investigations.

Many studies using ex vivo acute slices have shown that chronic visual deprivation leads to synaptic enrichment of AMPARs in V1 L2/3 neurons through synaptic scaling mechanisms (*Goel and Lee, 2007*; *He et al., 2012*). These reports used whole-cell recordings to determine AMPAR-mediated miniature excitatory postsynaptic current (mEPSC) amplitude or biochemistry methods to measure synaptic AMPAR expression (*Goel et al., 2006*; *Goel and Lee, 2007*), which both reflect average AMPAR level of all synapses from a cell or a large number of cells in V1, and thus lacks crucial spatial information describing the dynamic behavior of individual synapses. Using in vivo imaging of fluorescently labeled AMPARs, we were able to track individual spine changes before and after visual deprivation, thus providing unprecedented levels of spatiotemporal information regarding synaptic AMPAR dynamics. We determined that changes in spine sGluA1 expression induced by visual deprivation are highly heterogeneous in live animals and only a subset of spines undergoes potentiation. These potentiated synapses are spatially organized and tend to be located deep within L2/3 neurons, predominately on basal rather than apical dendrites, and this depth-dependent mechanism even applies to spines along the same dendrite, indicating that deprivation-induced changes are

compartment- and input-specific, as apical dendrites and basal dendrites of L2/3 neurons receive different inputs (*Ko et al., 2011*; *Makino and Komiyama, 2015*; *Petrus et al., 2015*; *Zhang et al., 2014*). In addition, it has been shown that the expression pattern of neurotransmitter receptors varies in different layers, and thus other extracellular factors like neuromodulators could also contribute to this depth-dependent plasticity (*Brombas et al., 2014*; *Ji et al., 2015*). Although homeostatic plasticity is generally thought to occur globally throughout the neuron, it can also occur locally (*Béïque et al., 2011*; *Turrigiano, 2012*). Our results reveal that the visual deprivation-induced homeostatic up-regulation of synaptic AMPARs is synapse-specific.

The increase in spine sGluA1 on the apical dendrites of L2/3 neurons occurred after 7 days of visual deprivation while in V1 region from the acute brain slice we saw a significant elevation of synaptic GluA1 after 2 days of deprivation. That discrepancy led us to investigate the changes happening on the basal dendrites of L2/3 neurons. Indeed, we observed that the amount of synaptic sGluA1 on the basal dendrites of L2/3 neurons was significantly increased after one day of deprivation and remained elevated afterwards. Therefore, the increase observed after 2 days ex vivo is primarily caused by the increase of synaptic sGluA1 on basal dendrites of L2/3 neurons. As far as we know, our study is the first report to show that synaptic sGluA1 on the apical dendrites of L2/3 neurons decreases first following visual deprivation. The dendrites that show the initial decrease are located in the superficial region and they probably receive top-down inputs from regions like retrosplenial cortex and cingulate (*Makino and Komiyama, 2015*; *Roth et al., 2016*; *Zhang et al., 2014*). Visual deprivation may induce LTD that results in reduced synaptic AMPAR level in those connections. Extracellular factors like neuromodulators could also contribute to this (*Brombas et al., 2014*; *Ji et al., 2015*).

Visual-deprivation-induced homeostatic plasticity has been reported to be lamina-specific and age-dependent in V1. For instance, L4 neurons have an early critical period from P16 to P21 during which visual loss homeostatically up-regulates excitatory synaptic transmission (*Desai et al., 2002*). In L2/3, homeostatic plasticity is expressed after P21 and persists into adulthood (*Goel and Lee, 2007*). In L6, dark exposure initiated before but not after P21 increases average amplitude of mEPSC (*Petrus et al., 2011*). Regarding L5 neurons, visual deprivation induces an increase in spine size and mEPSC amplitude of L5 neurons in adult mice (*Barnes et al., 2017*; *Keck et al., 2013*). However, in the same study they also found that such deprivation did not cause a homeostatic response of L2/3 neurons, which is contrary to other ex vivo studies (*Barnes et al., 2015*; *Goel and Lee, 2007*). These discrepancies could result from different deprivation paradigms being used, such as monocular or binocular enucleation, dark exposure, eyelid suture (*Whitt et al., 2014*). Notably, a study has carefully and systematically investigated how varying visual deprivation paradigms affect plasticity in V1 and demonstrates that a complete loss of visually driven cortical activity is required to elicit homeostatic plasticity in V1 L2/3 pyramidal neurons (*He et al., 2012*). Here, we used binocular enucleation to completely deprive visual inputs and showed that this paradigm successfully induces up-regulation of synaptic sGluA1 in V1 L2/3 neurons, which is consistent with previous ex vivo studies. However, we did not see any increase of synaptic sGluA1 on apical dendrites of L5 neurons following the same deprivation in adult animals. The increase in mEPSC amplitude of L5 neurons induced by visual deprivation could be contributed by the basal dendrites, as here we only imaged the apical dendrites of L5 neurons. Due to their deep locations, we are not able to image the basal dendrites of L5 neurons with our two-photon system. In addition, it has been shown that in the barrel cortex there is distinct plasticity triggered by sensory changes in specific subtypes of L5 neurons (*Greenhill et al., 2015*; *Holtmaat et al., 2006*). In the primary visual cortex, we also observed two populations of L5 cells showing different responses to visual deprivation, with one increasing sGluA1 and the other one not. Future studies are necessary to examine whether these two different responses result from distinct subtypes of the cells. Also, it will be interesting to know whether the homeostatic plasticity in L5 neurons is development-dependent or not.

Emerging evidence has shown that loss of a sensory input leads to widespread changes across brain areas, including the deprived sensory cortex and spared sensory cortices (*Ibrahim et al., 2016*; *Lee and Whitt, 2015*; *Petrus et al., 2015*). For example, visual deprivation not only induces homeostatic plasticity in primary visual cortex, but also produces compensatory changes in other spared sensory cortices, such as somatosensory cortex and auditory cortex, with decreases of mEPSC amplitude in L2/3 pyramidal neurons (*Lee and Whitt, 2015*; *Petrus et al., 2015*). However, how visual deprivation alters non-V1 visual cortex remains unknown. In the study, we showed that loss of vision

caused a reduction in synaptic sGluA1 of L2/3 neurons within non-V1 visual cortex. Notably, unlike somatosensory and auditory cortices, non-V1 visual cortex exhibited a much earlier decrease. Therefore, the adaptation of brain circuits within visual cortex is different from those in spared sensory cortices.

Overall, by tracking the strength of individual synapses with longitudinal imaging of AMPAR dynamics in living animals during visual deprivation, we reveal that synaptic inputs to distinct cortical layers are differentially modulated in response to sensory experience. Our study supports the notion that the adult brain remains remarkably malleable and is continuously reshaped by experience.

# Materials and methods

All experimental protocols were conducted according to the National Institutes of Health guidelines for animal research and were approved by the Animal Care and Use Committee at Johns Hopkins University School of Medicine.

**Key resources table**

| Reagent type (species) or resource | Designation | Source or reference | Identifiers | Additional information |
|---|---|---|---|---|
| Antibody | Anti-dsRed2 (rabbit polyclonal) | Clontech | Cat# 632496, RRID:AB_10013483 | 1:1000 |
| Antibody | Anti-GFP (chicken polyclonal) | Abcam | Cat# ab13970, RRID:AB_300798 | 1:1000 |
| Antibody | Alexa Fluor 568 goat anti-rabbit | Thermo Fisher Scientific | Cat# A-11011, RRID:AB_143157 | 1:500 |
| Antibody | Alexa Fluor 488 goat anti-chicken | Thermo Fisher Scientific | Cat# A-11039, RRID:AB_2534096 | 1:500 |
| Antibody | Anti-GluA1 (JH4294) | (*Oku and Huganir, 2013*) | N/A | |
| Antibody | Anti-GluN1 (JH2590) | (*Liao et al., 1999*) | N/A | |
| Chemical compound, drug | MNI-caged-L-glutamate | Tocris | Cat#0218, N/A | |
| Strain, strain background | Mouse: WT C57BL/6N | Charles River | Strain #027, RRID:IMSR_CRL:027 | |
| Strain, strain background | *Nestin-Grip1* fl/fl mouse | (*Takamiya et al., 2008*) | N/A | |
| Recombinant DNA reagent | SEP-GluA1 | (*Zhang et al., 2015*) | N/A | |
| Recombinant DNA reagent | Myc-GluA2 | (*Zhang et al., 2015*) | N/A | |
| Recombinant DNA reagent | DsRed2 | Clontech | Cat# 632405 | |
| Software, algorithms | MATLAB | Mathworks | https://www.mathworks.com, RRID:SCR_001622 | |
| Software, algorithms | ScanImage | (*Pologruto et al., 2003*) | https://www.vidriotechnologies.com, RRID:SCR_014307 | |
| Software, algorithms | IGOR Pro | WaveMetrics | https://www.wavemetrics.com/products/igorpro, RRID:SCR_000325 | |
| Software, algorithms | MapManager | (*Zhang et al., 2015*) | https://mapmanager.net/ | |
| Software, algorithms | ImageJ | (*Schneider et al., 2012*) | https://imageJ.net/, RRID:SCR_003070 | |

## In utero electroporation

Progenitor cells in the visual cortex were transfected with SEP-GluA1, myc-GluA2, and dsRed2 (4:2:1 ratio, respectively) by in utero electroporation of E15 and E14 embryos to label L2/3 and L5

pyramidal neurons, respectively as previously described (*Saito and Nakatsuji, 2001*; *Zhang et al., 2015*). Timed pregnant C57BL/6N mice (Charles River) or *Grip1* conditional knockout mice were anesthetized with Avertin (0.02 ml/mg). Approximately 0.5–1 µl of DNA solution containing fast green was pressure injected into the left lateral ventricle of each embryo through a pulled-glass pipette. The head of each embryo was placed between two forceps-type electrodes. The anode contacted the prefrontal side of left hemisphere and the cathode faced the occipital side of the injected ventricle to target the visual cortex. Five pulses of 35 V for L2/3 or 30 V for L5 (50 ms duration, 1 Hz) were delivered through a square wave electroporator (CUY21, BEX Co, LTD., Japan).

## Craniotomy

Electroporated animals were subsequently implanted with a 3 × 3 mm cranial window over the visual cortex region at the age of two months. Mice were anesthetized with Avertin and the skull was sealed using dental cement. A metal head bar was attached to the skull during the surgery to allow head fixation for future two-photon imaging. Carprofen (4–5 mg/kg) and Dexamethasone (4–5 mg/kg) were injected to reduce pain and inflammation during the surgery. The antibiotics sulfamethoxazole (1 mg/ml) and trimethoprim (0.2 mg/ml) were chronically administered in the drinking water, and the animals were housed individually after surgery.

## Optical intrinsic imaging

One week after the cranial window surgery, optical intrinsic imaging was performed as previously described (*Kalatsky and Stryker, 2003*; *Nauhaus and Ringach, 2007*). Mice were anesthetized and maintained on 0.75% isoflurane supplemented by xylazine (13 mg/kg). Drifting bar stimuli (vertically or horizontally) were displayed on a gamma-corrected LCD screen, which was placed 20 cm away from the right eye, covering the majority of unilateral visual space. The bar stimulus drifted 10 times along each cardinal axis. Spherical correction was applied to the stimulus to define eccentricity in spherical coordinates. Optical images of the visual cortex were acquired at 30 Hz using a CMOS (Complementary Metal-Oxide-Semiconductor) camera (FLIR GS3-U3-23S6M-C) under red LED light (630 nm) with a 2.5×/0.075 numerical aperture (NA) Zeiss objective. Multiframe image stacks were averaged across 30 trials. Next the images were Gaussian filtered ($\sigma$ = 2 pixels) and baseline was subtracted. V1 was delineated by a strong visual response with orthogonal retinotopy contours and the appropriate visual field sign (*Garrett et al., 2014*). Non-V1 is the region outside of V1 but still responding to visual stimulation.

## Two-photon imaging

In vivo images were acquired of awake mice with a custom-built, two-photon laser-scanning microscope controlled by ScanImage (Vidrio, Ashburn, VA) written in MATLAB (*Pologruto et al., 2003*). Mice were habituated under the microscope for one hour per day starting at one week before the beginning of imaging and subsequently imaged over a period of 10 days. Apical or basal dendrites of L2/3 or L5 pyramidal neurons of mouse visual cortex were imaged using a 20×/1.0 NA water-immersion objective lens (Zeiss). SEP-GluA1 and dsRed2 were excited at 910 nm with a Ti:sapphire laser (Coherent) with 10 ~ 150 mW of power delivered to the back-aperture of the objective. Green and red fluorescence signals were acquired simultaneously and separated by a set of dichroic mirrors (MOM system, Sutter Instrument) and filters (ET525/50 m for green channel, ET605/70 m for red channel, Chroma). Image stacks were acquired at 1,024 × 1024 pixels with a voxel size of 0.12 µm in x and y and a z-step of 1 µm. Representative images shown in figures were masked based on dendritic dsRed2 signal, median filtered, and contrast enhanced.

## Binocular enucleation

Enucleation mice were shortly anesthetized with isoflorane vapor first and then both eyes were surgically removed (*Aerts et al., 2014*). Antibiotic ointment was applied and carprofen was administrated immediately after the enucleation. Control sham mice were given time-matched anesthesia, and received antibiotic ointment treatment and carprofen administration, but were not enucleated. Mice were then returned to their home cage and monitored daily to make sure there was no bleeding or infection.

## Neuronal culture and transfection

Rat embryonic (E18) hippocampal neurons were plated on poly-L-lysine coated tissue culture dishes at a density of 30,000 cells/cm$^2$ and grown in neurobasal media (Invitrogen) supplemented with 2% (vol/vol) B-27, 2 mM GlutaMAX, 50 U/mL PenStrep. Cultured neurons were fed once per week and used at DIV 18–21. 2–3 days before uncaging experiments, neurons were transfected with SEP-GluA1, myc-GluA2 and dsRed2 (4:2:1) using lipofectamine 2000 (Invitrogen) according to the manufacturer's instructions.

## Glutamate uncaging and voltage-clamp recordings

Cultured rat hippocampal neurons were imaged and recorded 2–3 days after transfection in a modified HEPES-based ACSF buffer (in mM): 140 NaCl, 5 KCl, 10 glucose, 10 HEPES, 2 CaCl$_2$, 1 MgCl$_2$, 0.001 TTX, and 2.5 MNI-caged-L-glutamate (Tocris), pH = 7.30 and 310–316 mOsm. Recordings were made at room temperature in recirculated ACSF (3 mL/min). Recording pipettes were fabricated (Flaming/Brown Micropipette Puller, Sutter Instruments) from borosilicate capillary glass (Sutter, 4–6 MΩ open-tip resistance) and filled with (in mM): 115 CsMeSO4, 2.8 NaCl, 5 TEACl, 0.4 EGTA, 20 HEPES, 3 MgATP, 0.5 NaGTP, 10 NaPhosphocreatine, and 2.5 QX-314, pH = 7.32 and 306 mOsm. Whole-cell voltage-clamp recordings were made using a MultiClamp 700B amplifier and Digidata 1440A digitizer (Axon Instruments). MNI-Glutamate (Tocris) was uncaged (1 ms pulse width, 0.2 Hz) with a two-photon laser (Spectra Physics, Santa Clara, CA) onto visually identified spines at a wavelength of 730 nm and a power of 20 mW at the objective back aperture. Glutamate uncaging position was calibrated and controlled using custom software developed in house (Scan Stim by Dr. Ingie Hong). To minimize the effect of electrotonic filtering caused by variable numbers of branch points between the site of dendritic uncaging and the somatic recording pipette, we uncaged exclusively on spines of secondary dendrites (4–8 spines/dendritic segment and 1–3 dendritic segments/neuron). The glutamate-uncaging-evoked excitatory postsynaptic current (uEPSC) was measured by pClamp (Axon Instruments) and synchronized triggering of the uncaging laser with voltage-clamp recordings. Representative images shown in figures were median filtered and contrast enhanced.

## Tissue collection

Mice were anesthetized by inhalation of isoflurane followed by immediate cervical dislocation. Brains were removed and primary visual cortices were dissected out on ice. For L1 micro-dissection, brains were sectioned in the coronal plane into 300 µm thick slices using a Vibratome (VT1200s, Leica) in ice-cold, oxygenated (95% O$_2$ and 5% CO$_2$) low-Ca$^{2+}$/high-Mg$^{2+}$ dissection buffer (in mM): 2.6 KCl, 1.25 NaH$_2$PO$_4$, 26 NaHCO$_3$, 211 sucrose, 11 glucose, 0.5 CaCl$_2$ and 7 MgCl$_2$. The slices were then stained with trypan blue dye for 30 s and washed with cold dissection buffer. L1 and L2-6 of primary visual cortex were dissected out on ice under a dissection microscope.

## PSD fractionation and western blot

Primary visual (V1) cortices from control and enucleation mice were homogenized on ice in homogenization buffer buffer (in mM): 320 sucrose, five sodium pyrophosphate, 1 EDTA, 10 HEPES pH 7.4, 0.0002 okadaic acid, protease inhibitor cocktail (Roche)] using a 26-gauge needle. Homogenate was then centrifuged at 800 × g for 10 min at 4°C to yield P1 (nuclear) and S1 (post-nuclear). S1 was centrifuged at 17,000 × g for 20 min to yield P2 (membrane) and S2 (cytosol). P2 was then resuspended in water adjusted to 4 mM HEPES pH 7.4 followed by 30 min' agitation at 4°C. Suspended P2 was centrifuged at 25,000 × g for 20 min at 4°C. The resulted pellet (synaptosome) was resuspended in 50 mM HEPES pH 7.4, mixed with an equal volume of 1% triton X-100, and agitated at 4°C for 10 min. The PSD fraction was generated by centrifugation at 25,000 x g for 20 min at 4°C. The PSD material was then resuspended in lysis buffer (PBS containing 50 mM NaF, 5 mM sodium pyrophosphate, 1% Nonidet P-40, 1% sodium deoxycholate, 0.02% SDS, 200 nM okadaic acid, and protease inhibitor cocktail). The protein concentration was determined by bicinchoninic acid assay (BCA) kit (Thermo Fisher) and material was analyzed by Western blot. The following antibodies were used: anti-GluA1 C-terminal polyclonal antibody (JH4294, made in house), anti-GluN1 polyclonal antibody (JH2590, made in house), anti-GRIP1 polyclonal antibody (Millipore).

## Immunohistochemistry

Mice were anesthetized with Avertin and transcardially perfused with 4% paraformaldehyde (PFA). The brain was removed and fixed in 4% PFA/PBS for 2 hr at room temperature. Brains were then sectioned in the coronal plane into 100 µm thick slices using a vibratome (VT-1000, Leica). Slices were first blocked in 1% BSA with 0.3% triton X-100 in PBS for 1 hr at room temperature and then incubated with primary antibodies overnight at 4°C followed by incubation with secondary antibodies for 2 hr at room temperature. Slices were mounted in PermaFluor mounting medium (Thermo Scientific) and tiled z stack images were obtained using a laser scanning confocal microscope (Zeiss LSM510). The following primary antibodies were used: rabbit anti-dsRed2 (1:1000, Clontech) and chicken anti-GFP (1:1000, Abcam). The following secondary antibodies were used: Alexa Fluor 568 goat anti-rabbit (1:500 Thermo Fisher Scientific) and Alexa Fluor 488 goat anti-chicken (1:500 Thermo Fisher Scientific).

## Fluorescence intensity analysis

Signal intensity in spines was analyzed using a custom-written software MapManager (https://map-manager.net) in Igor Pro as previously described (*Zhang et al., 2015*). Briefly, each spine was assigned two regions of interest (ROIs) with a $spine_{ROI}$ enclosed the spine head and a $shaft_{ROI}$ enclosing the dendritic shaft adjacent to that spine. A $background_{ROI}$ (same shape and number of pixels as the $spine_{ROI}$ and $shaft_{ROI}$) was translated in x/y to a nearby region of the image that was representative of the background fluorescence. Intensity of SEP-GluA1 represents surface sGluA1 expression as SEP signal is pH-dependent, whereby acidic intracellular environments quenches the fluorescence. To compare intensity values between imaging sessions, the background subtracted $spine_{ROI}$ from either the SEP-GluA1 or dsRed channel was normalized to background subtracted the dsRed signal on the adjacent dendritic $shaft_{ROI}$. Further forms of normalizations were performed for different analyses as described in the following paragraph.

In *Figure 1G–I*, spine surface SEP-GluA1 (sGluA1) level of each spine was normalized to its individual day one intensity and the geometric mean of spine change per dendrite was calculated.

In *Figure 2B,C,H*, relative spine sGluA1 level was calculated by normalizing to the average level of the dendrite where the spine was located. In *Figure 2F*, *Figure 2—figure supplement 1A,B*, the intensities of newly formed spines were measured when the spines were first detected; the intensities of eliminated spines were measured when the spines were last detected. In *Figure 2H*, *Figure 2—figure supplement 1D*, relative spine size (y axis) or sGluA1 (x axis) was calculated by normalizing to the average level of the dendrite where the spine was located. In *Figure 2I*, *Figure 2—figure supplement 1E*, spine size (y axis) or sGluA1 (x axis) daily change was calculated by normalization to the level on the previous day.

In *Figure 3B,C,E,F*; *Figure 4D,F*; *Figure 5F,G*; *Figure 6F, H*; *Figure 7E*; *Figure 3—figure supplement 1A,C*; *Figure 4—figure supplement 1C*; *Figure 5—figure supplement 1B, C, K*; *Figure 6—figure supplement 1B,C,E,F*; *Figure 7—figure supplement 1A,B*, individual spine sGluA1/ size levels were normalized to the average of three baselines (BL1-BL3) and the geometric mean of spine change per dendrite was calculated. In *Figure 5H*, the change ratio on each imaging session was calculated by normalizing changes of spine sGluA1 level on basal dendrites to the change in spine sGluA1 expression on apical dendrites of the same neuron. In *Figure 4D,E*; *Figure 6H*, dendrites were defined as decrease dendrites if they had a significant decrease of at least 18% (standard deviation of the percent change in the sham group) in spine sGluA1 at VD1; Otherwise, they were defined as 'no decrease' dendrites. In *Figure 4—figure supplement 1C*, spines were defined as decrease spines if they had a significant decrease of at least 43.5% (standard deviation of the percent change in the sham group) in spine sGluA1 at VD1; Otherwise, they were defined as 'no decrease' spines. In *Figure 6H*, cells were defined as decrease cells if their apical dendrites had a significant decrease of at least 9.8% (standard deviation of the percent change in sham group) in spine sGluA1 at VD1; Otherwise, they were defined as 'no decrease' cells.

In *Figure 5—figure supplement 1E*; *Figure 6—figure supplement 1H*, the depth of dendrite was calculated by averaging all spines on the dendrite. In each imaging ROI, the relative depth of dendrites was calculated as the Z distance relative to the most superficial dendrite (depth = 0) and the deepest dendrite (depth = 1). In *Figure 5—figure supplement 1F–J*, apical dendrites and basal dendrites were combined together. In *Figure 6—figure supplement 1I,J*, apical dendrites were

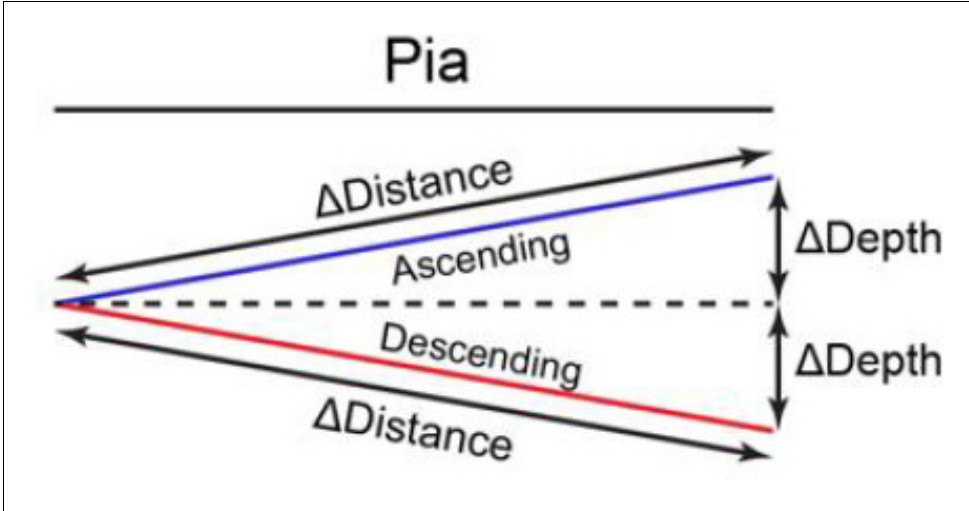

**Scheme 1.** Definitions of ascending dendrites and descending dendrites.

analyzed. As shown in *Figure 5—figure supplement 1D*, for depth analysis, spine depth from the pia was calculated as the z distance relative to the most superficial spine (depth = 0) and the deepest spine (depth = 1) on the same dendrite. For distance analysis, on each dendrite, spine distance was calculated relative to the most proximal spine to the branch point (defined as 0) and the most distal spine (defined as 1). Spine sGluA1 change was calculated by normalization to the average sGluA1 change of all persistent spines on the same dendrite. The dendrites were defined as 'ascending' dendrites if the dendrites were extending towards pia and $\Delta$Depth / $\Delta$Distance was larger than 0.1 (*Scheme 1*). The dendrites were defined as 'descending' dendrites if the dendrites were extending away from pia and $\Delta$Depth / $\Delta$Distance was larger than 0.1 (*Scheme 1*). For all imaging analysis, the averages were calculated per dendrite.

## Statistical analysis

All statistical analyses were performed in GraphPad Prism 7. Data distribution was tested for normality (Shapiro-Wilk test) and then comparisons were made using parametric or non-parametric tests, as appropriate. Statistical significance was determined by Student's t-test, one-sample t-test, Kolmogorov-Smirnov test, one-way or two-way ANOVA with Bonferroni post hoc test, Friedman test with Dunn's post hoc test as indicated in the figure legends.

## Acknowledgements

We would like to thank Dr. Ingie Hong, Elena Lopez-Ortega for insightful discussions and technical guidance, and other members of Huganir lab for their advice and support. This work was supported by National Institute of Health Grants R01NS036715 and P50MH100024 (to RLH) and AHA SDG grant 16SDG27130006 (to RHC).

## Additional information

### Funding

| Funder | Grant reference number | Author |
| --- | --- | --- |
| National Institute of Neurological Disorders and Stroke | R01NS036715 | Richard L Huganir |
| National Institute of Mental Health | P50MH100024 | Richard L Huganir |
| American Heart Association | 16SDG27130006 | Robert H. Cudmore |

The funders had no role in study design, data collection and interpretation, or the decision to submit the work for publication.

### Author contributions
Han L Tan, Conceptualization, Data curation, Formal analysis, Investigation, Visualization, Methodology; Richard H Roth, Data curation, Investigation; Austin R Graves, Data curation; Robert H Cudmore, Resources, Software; Richard L Huganir, Conceptualization, Resources, Supervision, Funding acquisition, Methodology, Project administration

### Author ORCIDs
Han L Tan (ID) https://orcid.org/0000-0001-5163-7720
Richard H Roth (ID) https://orcid.org/0000-0002-6855-999X
Robert H Cudmore (ID) https://orcid.org/0000-0002-0440-1704
Richard L Huganir (ID) https://orcid.org/0000-0001-9783-5183

### Ethics
Animal experimentation: This study was performed in strict accordance with the recommendations in the Guide for the Care and Use of Laboratory Animals of the National Institutes of Health. All of the animals were handled according to approved institutional animal care and use committee (IACUC) protocol # MO17M358 of Johns Hopkins University School of Medicine. The protocol was approved by the Animal Care and Use Committee at Johns Hopkins University School of Medicine.

### Decision letter and Author response
Decision letter https://doi.org/10.7554/eLife.52420.sa1
Author response https://doi.org/10.7554/eLife.52420.sa2

## Additional files

### Supplementary files
• Supplementary file 1. Two-photon imaging of apical dendrites from V1 L2/3 neurons. Channel 1 (SEP-GluA1 signal).

• Supplementary file 2. Two-photon imaging of apical dendrites from V1 L2/3 neurons. Channel 2 (dsRed2 signal).

• Supplementary file 3. Source data of spine sGluA1.

• Transparent reporting form

### Data availability
All data generated or analyzed (changes of fluorescence over time) during this study are included in the manuscript and supporting files. We also provide one set of raw imaging data (Supplementary files 1 and 2). Due to the large volume of imaging data sets, all raw imaging data are on a local secure server from Huganir lab and will be available upon request. We provide the raw source data generated by our analysis software from the raw images (Supplementary file 3). Signal intensity in spines was analyzed using a custom-written software MapManager (https://mapmanager.net) in Igor Pro. Full documentation and source code download is available at https://github.com/mapmanager.

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
