## [Decision Letter]

**Acceptance summary:**

Understanding synaptic plasticity is a major goal of developmental neuroscience. Tan et al. have demonstrated techniques for following hundreds of identified dendritic spines in specific neuron classes over the critical period for ocular dominance of mouse visual cortex, with and without visual deprivation by enucleation. The authors demonstrate that spine dynamics depend on the cell type and dendrite type (apical vs. basal), suggesting that the mechanisms of synaptic refinement are not uniform but vary across specific elements and sub-elements of neural circuits.

**Decision letter after peer review:**

Thank you for submitting your article "Lamina-specific AMPA receptor dynamics following visual deprivation in vivo" for consideration by *eLife*. Your article has been reviewed by three peer reviewers, one of whom is a member of our Board of Reviewing Editors, and the evaluation has been overseen by Ronald Calabrese as the Senior Editor. The following individual involved in review of your submission has agreed to reveal their identity: Corette J. Wierenga (Reviewer #2).

The reviewers have discussed the reviews with one another and the Reviewing Editor has drafted this decision to help you prepare a revised submission.

Summary:

The reviewers were united in admiring the quality of the raw data collected and its contribution to our understanding how synaptic spines are influenced by plasticity. At the same time, many concerns were raised about the interpretations and conclusions drawn in the text. The reviewers were united in suggesting a major revision of the paper for consideration for publication in *eLife*.

Essential revisions:

1) "Hot" and "cold" dendrites: The authors cite the existence of dendrites that exhibit "hot" (most synapses changed) and "cold" (most synapses did not change) dynamics. How do we know that spines with changes are more likely than chance to be located next to each other? There is a certain probability that such a situation could arise from chance, and no effort was made to see if the rate of occurrence of these hot and cold dendrites was greater or less than expected by a random arrangement of spines onto the dendrites (which could be examined by making 1000s of surrogate "location-shuffled" datasets and calculating percentiles). We appreciate that the authors have spent a long time looking at the data and may have some reason to believe that the hot and cold dendrites might be a real phenomenon, but at present there is no evidence in the paper that convince us it is real.

2) Cell-type-specificity vs. dendrite-specificity: The authors claim that layer 5 cells exhibit different plasticity than layer 2/3 neurons. We do understand that the plasticity was different in the observed layer 2/3 and layer 5 spines. However, there seem to be other equally parsimonious explanations. For instance, the following alternative hypothesis: basal dendrites exhibit prolonged homeostatic plasticity while apical dendrites and their branches do not. I imagine that layer 5 spines were only examined in apical dendrites, and so this possibility could not be rejected. Instead of claiming that cell types exhibit different plasticity, a broader set of possibilities should be enumerated in the Abstract, Results, and Discussion. Further, can the authors link their findings to possible VD-induced changes in V1 circuitry? Which synapses are changing and which not? Do L4 inputs target mostly apical or basal dendrites of L2/3 cells? Why would L5 cells not be affected?

3) The observed increase in sGluA1 only occurs after 7 days, while the increase is already observed after 2 days in vitro. Also the initial decrease in sGluA1 observed in vivo has never been reported in vitro (as far as we know). What do the author think is happening at dendrites that show the initial decrease? The authors should discuss this remarkable time course.

4) While GluA1 is correlated with synaptic strength (Figure 1D), the two quantities are not equal; there is considerable variation around the linear correlation. Please rephrase.

5) No account is taken of the well documented differences in homeostatic response and mechanism involved in subtypes of layer 5 neuron.

6) The GRIP KO does show an increase in GluA1 between time VD3 and VD7 points in complete contrast to the conclusion in the text.

7) Unfortunately, the paper does not pay much attention to the quite extensive literature on the mechanism of homeostatic plasticity, nor a good deal of literature conducted on GluA1 knockouts in visual cortex. Neither does it frame the experiments within the appropriate context, because it equates monocular deprivation, dark exposure and eye enucleation experiments within and outside the critical period for development of binocular vision, and thereby cloaks differences between the methods under the misleading term (here) of "sensory deprivation".

Reviewer #2:

In this manuscript the authors have performed longitudinal two-photon imaging to follow changes in spine size and AMPA receptor content in vivo during visual deprivation. They provide evidence that not all synapses respond in a similar manner and interpret this to reflect a depth-dependent plasticity. These new in vivo data are very valuable and the findings are intriguing. I have listed some suggestions to further improve manuscript below.

1) The authors claim that the synaptic changes depend on cortical depth, but I find this a rather vague concept, which is not much discussed. Can the authors speculate if this is linked to a specific input or to a extracellular factor? Can the authors link their findings to possible VD-induced changes in V1 circuitry? Which synapses are changing and which not? Do L4 inputs target mostly apical or basal dendrites of L2/3 cells? Why would L5 cells not be affected?

2) The authors make a strong claim that VD-induced synaptic changes do not occur in L5 (Discussion, sixth paragraph) and that L5 dendrites, in contrast to L2/3 cells, do not show depth-dependent changes after deprivation (e.g. subsection “Lamina-specific increases of spine sGluA1 level after deprivation”, last paragraph). However, they only looked in a small subset of dendrites in L1 and they cannot exclude changes in basal dendrites or dendrites closer to the soma. Furthermore, they actually found a small decrease in sGluA1 in L5 cells after VD (Figure 7E). In addition, the effect in L2/3 as shown in Figure 5—figure supplement 1D is pretty subtle and the number of observations is a lot lower in Figure 6—figure supplement 1I. The most important finding in L2/3, the difference between basal and apical dendrites (or distal/proximal dendrites), was not assessed in L5 cells. I therefore think this claim is too strong and I would ask the authors to either provide additional evidence or to tone this statement down.

Reviewer #3:

This is an interesting paper in many ways, extending the literature on imaging spines size longitudinally, to look at the role played by GluA1 and GRIP in the process of homeostatic response to eye enucleation. Several of the results are of great interest and well founded in the data, for example:

1) The difference in GluA1 dynamics and response to enucleation of apical versus basal dendrites of layer 2/3.

2) The heterogeneous response of (apical?) dendrites of layer 5 neurones.

However, in a number of places, the paper also includes interpretations of the data that are either not supported by the data at all, or comprise such minor components of the highly variable data that they are probably of little biological significance, even if true. There are numerous examples of the two types of issue, but in summary they include:

1) That GluA1 fluorescence does not equate to synaptic strength.

2) Purported difference in fluorescence by depth measurement along ascending or descending dendrites are based on data where depth is a tiny contributor to the large variance.

[Editors' note: further revisions were suggested prior to acceptance, as described below.]

Thank you for resubmitting your article "Lamina-specific AMPA receptor dynamics following visual deprivation in vivo" for consideration by *eLife*. Your article has been reviewed by two peer reviewers, one of whom is a member of our Board of Reviewing Editors, and the evaluation has been overseen by Ronald Calabrese as the Senior Editor. The following individual involved in review of your submission has agreed to reveal their identity: Corette J. Wierenga (Reviewer #2).

The reviewers have discussed the reviews with one another and the Reviewing Editor has drafted this decision to help you prepare a revised submission.

Summary and Essential revisions:

The new manuscript is improved, but reviewers were concerned that a few very weak trends were highlighted along with the manuscripts strongest findings. Please remove the part about hot dendrites, which were not upheld in statistical analysis, and move Figure 2J-K, Figure 5I-K to supplement or leave them out completely, and update the text to reflect the changes.

Reviewer #1:

The manuscript by Tan et al. is improved over the previous version. However, there is still one big problem.

The statistical shuffling analysis shows that the likelihood of observing the number of "hot" dendrites that were observed in these studies is about 30% (Figure 2—figure supplement 1A, the authors report 53.21% but adding 30% and 40% for 0 or 1 seems more like 70% total for 0 and 1), while the likelihood of observing 3 cold dendrites was very unlikely (3.82%). There is therefore no significant evidence for "hot" dendrites. The authors ought to remove such a reference. The case is better for cold dendrites; there is some evidence for a few cold dendrites. If it were me, I might remove this bit entirely because the numbers are small, but the authors can decide if they want to leave in the part about cold dendrites.

The other changes were satisfactory.

Reviewer #2:

The authors have addressed most of my comments in a positive manner. The manuscript has clearly improved and I thank the authors for their complete answer to my and the other reviewers' comments. I particularly appreciate the (perhaps somewhat speculative) discussion on how the depth-dependence of synaptic changes after binocular enucleation could reflect changes in specific inputs. This emphasizes the biological relevance of their findings.

I have the feeling that the manuscript would improve by a stronger focus on the strongest and most interesting results. In my opinion, the presentation of small effects and weak correlations of which the biological relevance are not directly clear is diluting the impact of the main results, which actually are of high interest. The authors may consider to move some of the minor findings (Figure 2J-K, Figure 5I-K) to the supplementary information (or leave them out completely).

---

## [Author Response]

Essential revisions:1) "Hot" and "cold" dendrites: The authors cite the existence of dendrites that exhibit "hot" (most synapses changed) and "cold" (most synapses did not change) dynamics. How do we know that spines with changes are more likely than chance to be located next to each other? There is a certain probability that such a situation could arise from chance, and no effort was made to see if the rate of occurrence of these hot and cold dendrites was greater or less than expected by a random arrangement of spines onto the dendrites (which could be examined by making 1000s of surrogate "location-shuffled" datasets and calculating percentiles). We appreciate that the authors have spent a long time looking at the data and may have some reason to believe that the hot and cold dendrites might be a real phenomenon, but at present there is no evidence in the paper that convince us it is real.

We have now examined the probability whether the observed “hot” or “cold” dendrites arise from chance or not by randomly shuffling the position of our spines 10,000 times. We found that both the number of “cold” dendrites and the sum of “hot” and” cold” dendrites (representing all dynamic dendrites) observed here were significantly greater than would occur by chance. The number of “hot” dendrites was also higher than the mean number of the shuffled data although it was not significant. That is probably due to the small number of dendrites we imaged. Nevertheless, these data clearly demonstrate that there are spatially clustered dendritic subregions where AMPARs are more readily down- or up- regulated. We have now added this comparison with shuffled data as Figure 2—figure supplement 1A.

2) Cell-type-specificity vs. dendrite-specificity: The authors claim that layer 5 cells exhibit different plasticity than layer 2/3 neurons. We do understand that the plasticity was different in the observed layer 2/3 and layer 5 spines. However, there seem to be other equally parsimonious explanations. For instance, the following alternative hypothesis: basal dendrites exhibit prolonged homeostatic plasticity while apical dendrites and their branches do not. I imagine that layer 5 spines were only examined in apical dendrites, and so this possibility could not be rejected. Instead of claiming that cell types exhibit different plasticity, a broader set of possibilities should be enumerated in the Abstract, Results, and Discussion. Further, can the authors link their findings to possible VD-induced changes in V1 circuitry? Which synapses are changing and which not? Do L4 inputs target mostly apical or basal dendrites of L2/3 cells? Why would L5 cells not be affected?

Due to their deep locations, we are not able to image the basal dendrites of L5 neurons with our two-photon system. We agree with the reviewer that we cannot exclude the possibility that the basal dendrites of layer 5 neurons could increase synaptic AMPARs following visual deprivation. We have changed our Abstract and Results, claiming that visual deprivation specifically increases synaptic AMPARs on apical dendrites of L2/3 neurons but not on apical dendrites of L5 neurons. In the Discussion, we also discuss the possibility that basal dendrites of L5 neurons could show synaptic enrichment of AMPARs after visual deprivation.

Our data show that the changes in synaptic sGluA1 induced by visual deprivation in V1 L2/3 neurons are significantly correlated with the depth (distance from the pia) of the dendrites or spines, wherein deep dendrites or spines are potentiated more than superficial ones (Figure 5; Figure 5—figure supplement 1). There are many possible mechanisms. First, as the reviewer mentioned, it could be input-specific, as apical dendrites and basal dendrites of L2/3 neurons receive different inputs. The basal dendrites primarily receive feedforward inputs from L4 and nearby L2/3 neurons (Ko et al., 2011; Lee et al., 2016; Park et al., 2019), while the apical dendrites receive feedback inputs from regions like retrosplenial cortex, cingulate, and thalamus (Makino and Komiyama, 2015; Roth et al., 2016; Zhang et al., 2014). In addition, the inhibitory projections to apical dendrites and basal dendrites of L2/3 neurons are distinct as well (Fino et al., 2013; Ma et al., 2014). Therefore, the depth-dependent changes could be driven by specific inputs. Second, as the reviewer also mentioned, it could be caused by extracellular neuromodulators, such as acetylcholine. For example, previous studies have shown that there is a distinct expression pattern of M2 muscarinic acetylcholine receptor in V1 (Ji et al., 2015). The expression is patchy in L1 where the apical dendrites of L2/3 neurons are located, but the expression of M2 muscarinic acetylcholine receptor in L2/3 where the basal dendrites of L2/3 neurons lie is less intense and uniform (Ji et al., 2015). It has been reported that acetylcholine can influence the excitability of interneurons in a cell-class dependent manner (Brombas et al., 2014). These distinct expression patterns of receptors in different depths could contribute to the depth-dependent changes we observed following visual deprivation. We have discussed these possible mechanisms in the Discussion.

Based on our results and previous literature, we think the feedforward inputs from L4 onto the basal dendrites of L2/3 neurons are potentiated in the beginning (one day after binocular enucleation) and show further potentiation afterwards. The top-down inputs from other regions onto the apical dendrites of L2/3 neurons are weakened first (one day after deprivation), but then recover and eventually undergo potentiation with prolonged deprivation (7 days).

In contrast to V1 L2/3 neurons, we did not see a significant increase of synaptic sGluA1 on apical dendrites of V1 L5 neurons after visual deprivation (Figure 6E, F). Many factors could account for this difference. First, L5 neurons and L2/3 neurons receive very different inputs. The canonical cortical microcircuit in V1 is that thalamic input drives activity in a feedforward and sequential fashion from L4 to L2/3 to L5 and out to other regions although numerous examples of alternate connections exist (Adesnik and Naka, 2018). L5 neurons are considered as one of the main integrators in the cortical column as their dendrites span all cortical layers and thus receive inputs from all layers (Briggs and Callaway, 2005). Further, L5 and L2/3 neurons differ in their dendritic arborization (Rojo et al., 2016; Spruston, 2008). L2/3 neurons have more confined dendritic trees compared to L5 neurons and apical dendrites of L5 neurons extend a greater distance than those of L2/3 neurons to reach the pia surface (Spruston, 2008). Indeed, there have been many studies demonstrating that L2/3 neurons and L5 neurons behave or function differently in response to changes in experience, such as whisker trimming, auditory stimulus, and motor learning (Holtmaat et al., 2006; Holtmaat et al., 2005; Sakata and Harris, 2009; Tjia et al., 2017).

We have now expanded our Discussion to include these circuit and cell-type specific effects of visual deprivation.

*3) The observed increase in sGluA1 only occurs after 7 days, while the increase is already observed after 2 days* in vitro*. Also the initial decrease in sGluA1 observed* in vivo *has never been reported* in vitro *(as far as we know). What do the author think is happening at dendrites that show the initial decrease? The authors should discuss this remarkable time course.*

The increase in spine sGluA1 on the apical dendrites of L2/3 neurons occurred after 7 days of visual deprivation while in V1 region from the acute brain slice we saw a significant elevation of synaptic GluA1 after 2 days of deprivation (Figure 3B; Figure 5A, B). That discrepancy led us to investigate the changes happening on the basal dendrites of L2/3 neurons. Indeed, we observed that the amount of synaptic sGluA1 on the basal dendrites of L2/3 neurons was significantly increased after one day of deprivation and remained elevated afterwards (Figure 5F-H). Therefore, the increase observed after 2 days in vitro is primarily caused by the increase of synaptic sGluA1 on basal dendrites of L2/3 neurons.

As far as we know, our study is the first report to show that synaptic sGluA1 on the apical dendrites of L2/3 neurons decreases first following visual deprivation and we confirmed that in vivo image data with in vitro biochemical experiments by micro-dissecting out L1 of V1 region (Figure 5C). This reduction only occurs on the apical dendrites of L2/3 neurons but not on their basal dendrites. Previous studies using whole-cell recordings or biochemical methods to examine L2/3 neuron or the whole V1 region are not able to detect that since they measure the average AMPAR level of all synapses from a cell or a larger number of cells in V1. This also highlights the advantage of our in vivo imaging technique that allows us to track individual spine changes during visual deprivation and provides unprecedented levels of spatiotemporal information regarding synaptic AMPAR dynamics.

The dendrites that show the initial decrease are located in the superficial region and they probably receive top-down inputs from regions like retrosplenial cortex and cingulate as we mentioned above. Visual deprivation may induce LTD that results in reduced synaptic AMPAR level in those connections. Extracellular factors like acetylcholine could also contribute to this.

We have now expanded our Discussion to include this remarkable time course.

4) While GluA1 is correlated with synaptic strength (Figure 1D), the two quantities are not equal; there is considerable variation around the linear correlation. Please rephrase.

We agree that spine sGluA1 level does not equate to synaptic strength despite the strong positive correlation detected. We have rephrased the sentences and claim that we are measuring spine sGluA1 level not synaptic strength.

5) No account is taken of the well documented differences in homeostatic response and mechanism involved in subtypes of layer 5 neuron.

We have reanalyzed our data based on the reviewer’s suggestion to look at different populations of layer 5 neurons. L5 basal dendrites are too deep to image in live animals with our two-photon system, and thus we were only able to image apical dendrites of L5 neurons that are located in L1. As a result, we do not have the whole structure of the cells we imaged. It is not reliable to tell which cells are thick tufted and which ones are thin tufted just based on their dendrites in L1. As an alternate way, we categorized the cells based on their responses following binocular enucleation: whether spine sGluA1 on apical dendrites decreased or not at day 1 after enucleation. We found that there are two populations of cells: one population showed decrease in the first few days following deprivation and then gradually recovered while the other group did not decrease at day 1 but exhibited a graduate increase despite that the increase is not significant (Figure 6H). As the reviewer mentioned, based on some previous studies (Greenhill et al., 2015; Holtmaat et al., 2006), the first population of cells might be RS thin tufted L5 cells and the second population might be IB thick tufted L5 cells. Further studies are necessary to confirm the identities of those cells. We have changed the Results and Discussion based on these new analyses.

6) The GRIP KO does show an increase in GluA1 between time VD3 and VD7 points in complete contrast to the conclusion in the text.

There is a recovery of spine GluA1 expression between VD3 and VD7. This recovery could be mediated through some compensatory regulations of other proteins involved in exocytosis of AMPARs, such as GRIP2 (Anggono and Huganir, 2012). We have rephrased our sentences and discussed that recovery. Nevertheless, there is no significant increase of spine sGluA1 following 7 days of enucleation compared to baseline level in GRIP1 KO animals, while in WT control animals synaptic sGluA1 is significantly increased after 7 days’ enucleation. We have compared the effects of genotypes (Figure 7—figure supplement 1) and there is a significant difference between WT control animals and GRIP1 KO animals. These results provide strong evidence that the increase of sGluA1 induced by enucleation is dependent on GRIP1.

7) Unfortunately, the paper does not pay much attention to the quite extensive literature on the mechanism of homeostatic plasticity, nor a good deal of literature conducted on GluA1 knockouts in visual cortex. Neither does it frame the experiments within the appropriate context, because it equates monocular deprivation, dark exposure and eye enucleation experiments within and outside the critical period for development of binocular vision, and thereby cloaks differences between the methods under the misleading term (here) of "sensory deprivation".

We have clarified our usage of “sensory deprivation” in the manuscript and added some discussion regarding the different effects and mechanisms of visual deprivation. Meanwhile, the homeostatic regulation in vivo in our manuscript refers to the homeostatic recovery of neuronal activity of pyramidal neurons in the visual cortex following visual deprivation in intact animals and this has been demonstrated by using either monocular eyelid suture or enucleation. (Barnes et al., 2015; Hengen et al., 2013; Hengen et al., 2016; Keck et al., 2013). Using chronic multielectrode recordings, Hengen et al., showed that the firing rate of L2/3 pyramidal neurons in monocular V1 region dropped first but recovered later following monocular lid suture (Hengen et al., 2013; Hengen et al., 2016). Similarly, the neuronal activity of L2/3 and L5 neurons in V1 measured by calcium imaging decreased after binocular lesions but restored later despite prolonged deprivation (Keck et al., 2013). The underlying mechanisms might be different, but both deprivation paradigms could induce homeostatic regulation of neuronal activity in the visual cortex. The molecular mechanisms underlying this recovery of neuronal activity in vivo remain largely unexplored, and here we examined AMPAR changes in live animals following binocular enucleation to provide molecular insights into this homeostatic plasticity.

As the reviewer pointed out, there are many differences between monocular visual deprivation and binocular deprivation, and different deprivation paradigms, such as enucleation, dark exposure, and eyelid suture, have distinct effects on the visual pathways and plasticity. Further, the plasticity induced by visual deprivation has been shown to be age-dependent (Whitt et al., 2014). We are aware of those differences and thus we confirmed some previous data in our hands with binocular enucleation before we went for further study. For example, when we found that binocular enucleation did not increase synaptic GluA1 on apical dendrites of V1 L2/3 neurons after 2 days. We confirmed that 2 days of binocular enucleation did increase synaptic GluA1 in V1 with ex vivo biochemical experiments, which led us to the hypothesis of depth-dependent changes. More importantly, in our study, all experiments were done under same conditions (same age of animals and same way to deprive the vision), and we did not combine different ways and mix the results. Therefore, we believe that our conclusions are well supported by our data.

Regarding the explanations of our data and how our data fit in with previous studies, we have discussed them in detail in the Discussion (fifth to eighth paragraphs). Differences between our study and previous findings could result from different paradigms used to deprive the vision or different ages of the animals being used. This is a very complex and still on-going topic and beyond the focus of our paper. We wished to study how visual deprivation through binocular enucleation affected real-time AMPAR dynamics in awake mice. Therefore, we did not spend much space in explaining the differences in the very beginning of the Introduction in order to avoid confusion and distraction from our main point. We have rephrased our sentences to make this clear. The GluA1 KO study (Ranson et al., 2013) is very interesting although the ocular dominance plasticity and homeostatic plasticity of neuronal activity are not exactly the same. It still implicates the importance of GluA1 in the visual plasticity and our observation of dynamic changes of GluA1-containing AMPARs during visual deprivation further supports that and provide more mechanistic details.

Reviewer #2:[…]1) The authors claim that the synaptic changes depend on cortical depth, but I find this a rather vague concept, which is not much discussed. Can the authors speculate if this is linked to a specific input or to a extracellular factor? Can the authors link their findings to possible VD-induced changes in V1 circuitry? Which synapses are changing and which not? Do L4 inputs target mostly apical or basal dendrites of L2/3 cells? Why would L5 cells not be affected?

Our data show that the changes in synaptic sGluA1 induced by visual deprivation in V1 L2/3 neurons are significantly correlated with the depth (distance from the pia) of the dendrites or spines, wherein deep dendrites or spines are potentiated more than superficial ones (Figure 5; Figure 5—figure supplement 1). There are many possible mechanisms. First, as the reviewer mentioned, it could be input-specific, as apical dendrites and basal dendrites of L2/3 neurons receive different inputs. The basal dendrites primarily receive feedforward inputs from L4 and nearby L2/3 neurons (Ko et al., 2011; Lee et al., 2016; Park et al., 2019), while the apical dendrites receive feedback inputs from regions like retrosplenial cortex, cingulate, and thalamus (Makino and Komiyama, 2015; Roth et al., 2016; Zhang et al., 2014). In addition, the inhibitory projections to apical dendrites and basal dendrites of L2/3 neurons are distinct as well (Fino et al., 2013; Ma et al., 2014). Therefore, the depth-dependent changes could be driven by specific inputs. Second, as the reviewer also mentioned, it could be caused by extracellular neuromodulators, such as acetylcholine. For example, previous studies have shown that there is a distinct expression pattern of M2 muscarinic acetylcholine receptor in V1 (Ji et al., 2015). The expression is patchy in L1 where the apical dendrites of L2/3 neurons are located, but the expression of M2 muscarinic acetylcholine receptor in L2/3 where the basal dendrites of L2/3 neurons lie is less intense and uniform (Ji et al., 2015). It has been reported that acetylcholine can influence the excitability of interneurons in a cell-class dependent manner (Brombas et al., 2014). These distinct expression patterns of receptors in different depths could contribute to the depth-dependent changes we observed following visual deprivation. We have discussed these possible mechanisms in the Discussion.

Based on our results and previous literature, we think the feedforward inputs from L4 onto the basal dendrites of L2/3 neurons are potentiated in the beginning (one day after binocular enucleation) and show further potentiation afterwards. The top-down inputs from other regions onto the apical dendrites of L2/3 neurons are weakened first (one day after deprivation), but then recover and eventually undergo potentiation with prolonged deprivation (7 days).

In contrast to V1 L2/3 neurons, we did not see a significant increase of synaptic sGluA1 on apical dendrites of V1 L5 neurons after visual deprivation (Figure 6E, F). Many factors could account for this difference. First, L5 neurons and L2/3 neurons receive very different inputs. The canonical cortical microcircuit in V1 is that thalamic input drives activity in a feedforward and sequential fashion from L4 to L2/3 to L5 and out to other regions although numerous examples of alternate connections exist (Adesnik and Naka, 2018). L5 neurons are considered as one of the main integrators in the cortical column as their dendrites span all cortical layers and thus receive inputs from all layers (Briggs and Callaway, 2005). Further, L5 and L2/3 neurons differ in their dendritic arborization (Rojo et al., 2016; Spruston, 2008). L2/3 neurons have more confined dendritic trees compared to L5 neurons and apical dendrites of L5 neurons extend a greater distance than those of L2/3 neurons to reach the pia surface (Spruston, 2008). Indeed, there have been many studies demonstrating that L2/3 neurons and L5 neurons behave or function differently in response to changes in experiences, such as whisker trimming, auditory stimulus, and motor learning (Holtmaat et al., 2006; Holtmaat et al., 2005; Sakata and Harris, 2009; Tjia et al., 2017).

We have now expanded our Discussion to include these circuit and cell-type specific effects of visual deprivation.

2) The authors make a strong claim that VD-induced synaptic changes do not occur in L5 (Discussion, sixth paragraph) and that L5 dendrites, in contrast to L2/3 cells, do not show depth-dependent changes after deprivation (e.g. subsection “Lamina-specific increases of spine sGluA1 level after deprivation”, last paragraph). However, they only looked in a small subset of dendrites in L1 and they cannot exclude changes in basal dendrites or dendrites closer to the soma. Furthermore, they actually found a small decrease in sGluA1 in L5 cells after VD (Figure 7E). In addition, the effect in L2/3 as shown in Figure 5—figure supplement 1D is pretty subtle and the number of observations is a lot lower in Figure 6—figure supplement 1I. The most important finding in L2/3, the difference between basal and apical dendrites (or distal/proximal dendrites), was not assessed in L5 cells. I therefore think this claim is too strong and I would ask the authors to either provide additional evidence or to tone this statement down.

Due to the deep location of L5 basal dendrites, we were not able to image those populations of dendrites with our two-photon imaging system. We agree that we cannot exclude the possibility that L5 basal dendrites could increase synaptic sGluA1 upon visual deprivation. We have changed our Abstract and Result, claiming that visual deprivation specifically increases synaptic AMPARs on the apical dendrites of L2/3 neurons but not on the apical dendrites of L5 neurons. In the Discussion, we also discuss the possibility that the basal dendrites of L5 neurons could show synaptic enrichment of AMPARs after visual deprivation.

Reviewer #3:[…]1) That GluA1 fluorescence does not equate to synaptic strength.

We agree with the reviewer that the GluA1 signal we imaged in the study does not equate to synaptic strength, but we think it is a good indicator or synaptic strength. Our data as well as many previous studies have observed a strong positive correlation between spine GluA1 level and synaptic strength (Makino and Malinow, 2011).

2) Purported difference in fluorescence by depth measurement along ascending or descending dendrites are based on data where depth is a tiny contributor to the large variance.

The biggest difference we saw is between apical dendrites and basal dendrites, which leads to the hypothesis that the increase induced by enucleation is depth-dependent. We thus examined whether this depth-dependent mechanism applied to dendrites in the same imaging region or even spines along the same dendrite. The correlation between the depth of the dendrites and the increases of sGuA1 induced by enucleation is small but significant (P = 0.006), and the correlation between the depth of the spines along the same dendrite and the increases of sGluA1 induced by enucleation is smaller but still significant (P = 0.0386) while in the control sham-surgery animals none of the correlations was significant. We acknowledge that the effects are small and the small r^2^ indicates that most likely they are not linearly correlated. However, we think these data are very interesting and could be biologically meaningful. Indeed, when we further categorized spines by dendritic compartment and then compared the sGluA1 intensity in the deepest 15% of spines with the most superficial 15% of spines along the same dendritic segment, we found that in both apical and basal dendrites spines positioned more deeply exhibited greater deprivation-induced changes in synaptic sGluA1 expression than spines more superficial poised along the same dendrite (Figure 5K). We could remove them if necessary.

[Editors' note: further revisions were suggested prior to acceptance, as described below.]

Reviewer #1:The manuscript by Tan et al. is improved over the previous version. However, there is still one big problem.The statistical shuffling analysis shows that the likelihood of observing the number of "hot" dendrites that were observed in these studies is about 30% (Figure 2—figure supplement 1A, the authors report 53.21% but adding 30% and 40% for 0 or 1 seems more like 70% total for 0 and 1), while the likelihood of observing 3 cold dendrites was very unlikely (3.82%). There is therefore no significant evidence for "hot" dendrites. The authors ought to remove such a reference. The case is better for cold dendrites; there is some evidence for a few cold dendrites. If it were me, I might remove this bit entirely because the numbers are small, but the authors can decide if they want to leave in the part about cold dendrites.

Thank you very much for your insightful comments. We have deleted this cluster result in the revised manuscript as suggested.

The other changes were satisfactory.Reviewer #2:The authors have addressed most of my comments in a positive manner. The manuscript has clearly improved and I thank the authors for their complete answer to my and the other reviewers' comments. I particularly appreciate the (perhaps somewhat speculative) discussion on how the depth-dependence of synaptic changes after binocular enucleation could reflect changes in specific inputs. This emphasizes the biological relevance of their findings.I have the feeling that the manuscript would improve by a stronger focus on the strongest and most interesting results. In my opinion, the presentation of small effects and weak correlations of which the biological relevance are not directly clear is diluting the impact of the main results, which actually are of high interest. The authors may consider to move some of the minor findings (Figure 2J-K, Figure 5I-K) to the supplementary information (or leave them out completely).

We thank the reviewer for the great comments and suggestions. We agree with the reviewer and have moved Figure 2J-K, Figure 5I-K to the supplement as suggested.